# Regulatory control of DNA end resection by Sae2 phosphorylation

Elda Cannavo[1], Dominic Johnson[2], Sara N. Andres[3,8], Vera M. Kissling[4], Julia K. Reinert [5,6], Valerie Garcia[2,9], Dorothy A. Erie[7], Daniel Hess [5], Nicolas H. Thomä [5], Radoslav I. Enchev[4], Matthias Peter [4], R. Scott Williams[3], Matt J. Neale[2] & Petr Cejka[1,4]

DNA end resection plays a critical function in DNA double-strand break repair pathway choice. Resected DNA ends are refractory to end-joining mechanisms and are instead channeled to homology-directed repair. Using biochemical, genetic, and imaging methods, we show that phosphorylation of *Saccharomyces cerevisiae* Sae2 controls its capacity to promote the Mre11-Rad50-Xrs2 (MRX) nuclease to initiate resection of blocked DNA ends by at least two distinct mechanisms. First, DNA damage and cell cycle-dependent phosphorylation leads to Sae2 tetramerization. Second, and independently, phosphorylation of the conserved C-terminal domain of Sae2 is a prerequisite for its physical interaction with Rad50, which is also crucial to promote the MRX endonuclease. The lack of this interaction explains the phenotype of *rad50S* mutants defective in the processing of Spo11-bound DNA ends during meiotic recombination. Our results define how phosphorylation controls the initiation of DNA end resection and therefore the choice between the key DNA double-strand break repair mechanisms.

---

[1] Faculty of Biomedical Sciences, Institute for Research in Biomedicine, Università della Svizzera italiana (USI), Bellinzona 6500, Switzerland. [2] Genome Damage and Stability Centre, School of Life Sciences, University of Sussex, Brighton BN1 9RH, UK. [3] Genome Integrity and Structural Biology Laboratory, National Institute of Environmental Health Sciences, Department of Health and Human Services, US National Institutes of Health, Research Triangle Park 27709-2233 NC, USA. [4] Department of Biology, Institute of Biochemistry, Eidgenössische Technische Hochschule (ETH), Zürich 8093, Switzerland. [5] Friedrich Miescher Institute for Biomedical Research, Basel 4058, Switzerland. [6] University of Basel, Basel 4056, Switzerland. [7] Department of Chemistry, Lineberger Comprehensive Cancer Center, University of North Carolina, Chapel Hill 27514 NC, USA. [8] Present address: Michael G. DeGroote Institute for Infectious Disease Research, Department of Biochemistry and Biomedical Sciences, McMaster University, Hamilton L8S4L8 ON, Canada. [9] Present address: Centre de Recherche en Cancérologie de Marseille (CRCM), Institut Paoli Calmettes, Inserm UMR1068, CNRS UMR7258, Aix Marseille Université, Marseille 13009, France. These authors contributed equally: Elda Cannavo, Dominic Johnson. Correspondence and requests for materials should be addressed to M.J.N. (email: m.neale@sussex.ac.uk) or to P.C. (email: petr.cejka@irb.usi.ch)

DNA double-strand breaks (DSBs) in vegetative cells arise from cellular exposure to radiation or chemicals, by enzymatic cleavage, or as a result of collapsed DNA replication forks. Cells possess two main mechanisms for DSB repair, either template-independent and often mutagenic end-joining pathways or largely accurate homology-directed recombinational repair[1]. The pathway choice must be carefully balanced to optimally maintain genome integrity at the lowest cost to mutagenesis. As the sister chromatid serves as a template for most recombinational repair events in vegetative cells, it is crucial to initiate recombination only during the S/G2 phase of the cell cycle, when such a template is available[1]. Illegitimate recombination in the absence of the correct repair template may lead to gross chromosomal rearrangements and promote cellular transformation in high eukaryotes[2].

The choice between end-joining and recombination-based mechanisms is governed by DNA end resection, which involves a selective degradation of the 5'-terminated DNA strand near a DSB[3]. DNA end resection generally inhibits end-joining and commits the DSB repair to homologous recombination. The resulting 3' overhangs are then coated by Rad51 to search for the homologous sequence[1]. At the molecular level, the initiation of end resection and hence recombination is regulated by phosphorylation of key resection factors[4,5]. This includes a cell cycle-dependent phosphorylation of *Saccharomyces cerevisiae* Sae2 or human CtIP by a cyclin-dependent protein kinase (CDK) at S267 of Sae2 or T847 of CtIP[6,7]. Indeed, the Sae2 S267E mutation mimicking constitutive phosphorylation partially bypasses the requirement for CDK in DSB resection[6]. Sae2/CtIP functions integrate with those of the Mre11-Rad50-Xrs2 (MRX, *S. cerevisiae*) or MRE11-RAD50-NBS1 (MRN, human) nuclease complex, but the nature of the interplay has remained elusive. Specifically, how phosphorylation of Sae2 at S267 and other, less defined, sites regulates DNA end resection on a mechanistic level has not been elucidated[8–12].

In meiosis, the Spo11 transesterase introduces in a programmed manner hundreds of DSBs during the prophase of the first meiotic division[13]. Unlike in vegetative cells where DSBs are generally detrimental and a threat to genome stability, meiotic DSBs are exclusively repaired by recombination using the homologous chromosome as a preferential template, which serves the purpose to generate diversity[1]. The resection and resulting recombinational repair of meiotic DSBs absolutely require Sae2 and MRX and their orthologs, because Spo11 remains covalently bound to the break ends and needs to be removed by MRX and Sae2[14–17]. This stands in contrast to resection in yeast vegetative cells, which can be partially MRX-Sae2 independent[18,19]. To this point, Rad50 separation of function alleles (*rad50S*) have been found, which prevent Spo11 removal in meiosis, but are less defective in vegetative cells[14,16,20–22]. Resection of all human DSBs, both in vegetative and meiotic cells, requires MRN and CtIP[23]. The processing of yeast meiotic DSBs thus serves as a model for events that require the Mre11/MRE11 nuclease.

We previously demonstrated that yeast Sae2 functions as a cofactor of the Mre11 nuclease to initiate resection of protein-bound DNA ends[24]. The MRX/N complex is a 3'→5' exonuclease that has the opposite polarity than that required for resection[25]. We and our colleagues showed that, in the presence of protein blocks located at or near DNA ends, which may include Ku, replication protein A (RPA), nucleosomes, Spo11 in meiosis, and possibly other yet unidentified factors[26–28], the 3'→5' exonuclease of MRX/N is inhibited. Instead, Sae2/CtIP stimulates the endonuclease of MRX/N to cleave preferentially the 5'-terminated DNA strand at DSB sites[24,29]. Here, using a combination of biochemical, genetic, and imaging tools, we define how phosphorylation regulates the capacity of Sae2 to initiate DNA end resection of blocked DNA ends. We show that phosphorylation at multiple sites promotes the formation of active Sae2 tetramers and additionally regulates a physical interaction with the Rad50 subunit of the MRX complex. These phosphorylation-dependent transitions turn Sae2 into a remarkably efficient activator of the Mre11 nuclease. Our results provide a mechanistic basis for the regulatory control of DSB repair pathway choice by Sae2 phosphorylation.

## Results

**Sae2 phosphorylation stimulates the MRX endonuclease.** Phosphorylated Sae2/CtIP promotes the endonuclease activity of Mre11 within the MRX/N complex[24,29]. Here we modified the Sae2 expression procedure by including phosphatase inhibitors in the *Spodoptera frugiperda* 9 culture media and cell extracts. This allowed us to obtain hyperphosphorylated recombinant Sae2 (Fig. 1a). The modified polypeptide exhibited electrophoretic mobility shift that was abolished upon treatment with λ phosphatase (Fig. 1a), which was reminiscent of its phosphorylation-dependent transitions observed in yeast cells extracts in vivo[8–10]. In contrast to our weakly phosphorylated previous preparation (denoted "Sae2" in Fig. 1, in blue)[24], the strongly phosphorylated Sae2 (hereafter denoted "pSae2," labeled red in Fig. 1), was up to ~10-fold more capable of stimulating the endonuclease of MRX using DNA substrates, where a single or both ends were blocked with streptavidin, serving as a protein block (Fig. 1b–f). Treatment of pSae2 with λ phosphatase dramatically reduced its capacity to promote the MRX endonuclease (Fig. 1e, f, labeled pSae2 λ in green). These findings allowed us to study the phosphorylation-dependent control of DNA end resection by pSae2 in a reconstituted system to define its underlying mechanism.

**Sae2 oligomerization is important for DNA end resection.** Hyperphosphorylated pSae2 had a lower DNA-binding affinity compared to the λ phosphatase-treated variant (Fig. 1g, Supplementary Fig. 1a). Furthermore, phosphorylation of Sae2 did not affect MRX's capacity to bind free or blocked DNA ends (Supplementary Fig. 1b–e). Therefore, the failure of non-phosphorylated Sae2 to stimulate the nuclease of MRX does not likely stem from a defect in DNA binding of Sae2 or the MRX–Sae2 complex.

The oligomeric state of biologically active Sae2 has been controversial[8,30]. To investigate the size distribution of our recombinant Sae2/pSae2 variants, we first employed size exclusion chromatography. Hyperphosphorylated pSae2, i.e., expressed with phosphatase inhibitors, eluted in an oligomeric state, spanning a range of peaks corresponding to a calibrated molecular weight of 400–600 kDa (Fig. 1h, Supplementary Fig. 2a). In contrast, most of Sae2 expressed without phosphatase inhibitors was present in a very large multimeric complex, equivalent to a globular species of several MDa, with only a small fraction being oligomeric, i.e., comparable to the size distribution of pSae2 (Fig. 1h). Dephosphorylation of pSae2 with λ phosphatase resulted in the formation of soluble multimers that shifted the polypeptide complex into the void volume of the size exclusion column (Fig. 1h, Supplementary Fig. 2a). The peak between 18 and 19 ml corresponds to λ phosphatase and not pSae2 (Fig. 1h). The capacity of pSae2 to stimulate MRX thus corresponded to the proportion of the polypeptide being an oligomer (Fig. 1f–h), suggesting that phosphorylation-dependent size distribution alters Sae2 function in DNA end resection.

To corroborate these data, we visualized the multimeric and oligomeric species by transmission electron microscopy (two-dimensional (2D) TEM). Indeed, the hypophosphorylated Sae2

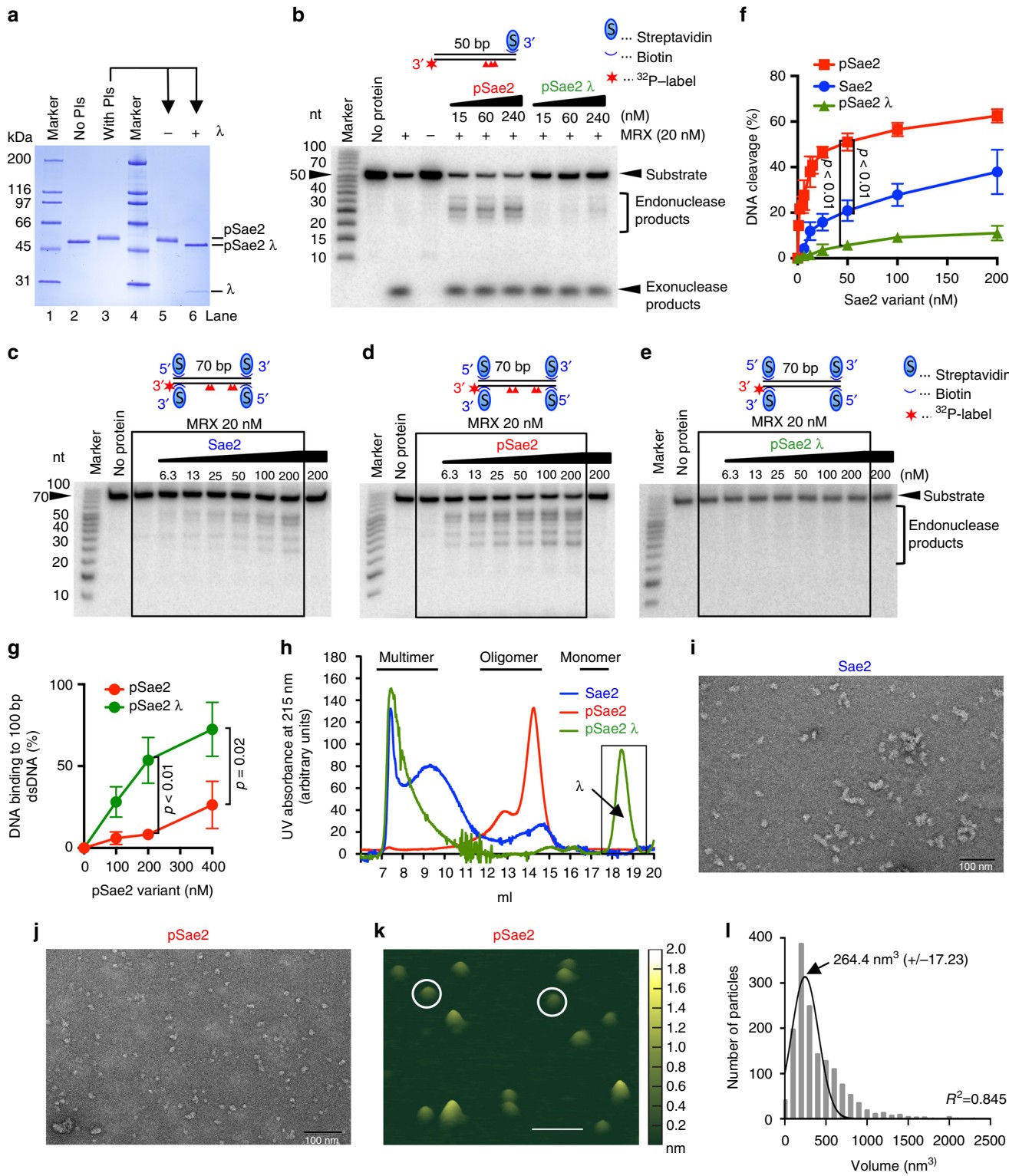

(corresponding to the blue species in Fig. 1f–h) was present in large multimeric complexes, while hyperphosphorylated pSae2 assemblies (corresponding to the red species in Fig. 1f–h), although heterogeneous, appeared much smaller (Fig. 1i, j and Supplementary Fig. 2b, c). As both size exclusion and 2D TEM only revealed an approximate average volume and molecular weight range, we used atomic force microscopy (AFM) to estimate the average number of subunits present in the active pSae2 complex in solution without DNA. Based on the estimate

of 17 Sae2 phosphorylation sites (see ref. [8] and below), the molecular weight of monomeric pSae2 is ~53.7 kDa. The volume analysis with AFM revealed a distribution of the pSae2 protomers with a peak corresponding to ~218.2 kDa based on a standard curve and thus a tetrameric form (Fig. 1k, l)[31], in agreement with results obtained with *Schizosaccharomyces pombe* Ctp1 and the N-terminal domain of human CtIP[32,33].

To further define the roles of pSae2 oligomerization on its capacity to promote MRX activity, we performed mutational

**Fig. 1** Phosphorylation of Sae2 regulates its capacity to promote the Mre11 endonuclease as well as its size distribution. **a** Recombinant Sae2 was expressed and purified either without (lane 2) or with (lane 3) phosphatase inhibitors (PIs). The phosphorylated pSae2 was mock-treated (lane 5) or treated with λ phosphatase (lane 6). **b** Phosphorylated (pSae2) and λ phosphatase-treated Sae2 (pSae2 λ) were used in nuclease assays with MRX and a DNA substrate with a single end blocked with streptavidin. Red triangles in the substrate cartoon indicate DNA cleavage positions. **c–e** Sae2 variant prepared without PIs (**c**, Sae2), with PIs (**d**, pSae2), and with PIs and then treated with λ phosphatase (**e**, pSae2 λ) were used in nuclease assays with MRX and a DNA substrate with both ends blocked with streptavidin. Red triangles in the substrate cartoon indicate DNA cleavage positions. **f** Quantitation of assays such as shown in **c–e**. Error bars, SEM, $n \geq 3$. **g** DNA-binding analysis of pSae2 untreated or treated with λ phosphatase, using 100-bp-long dsDNA as a substrate. Quantitation of experiments such as shown in Supplementary Fig. 1a. Error bars, SEM, $n = 3$. **h** Size exclusion chromatography analysis of Sae2 variants prepared without PIs (Sae2, blue), with PIs (pSae2, red), and Sae2 prepared with PIs and then treated with λ phosphatase (pSae2 λ, green). The black rectangle indicates the position of the λ phosphatase peak. **i, j** Representative transmission electron microscopic images of Sae2 prepared without (Sae2, 800 nM, **i**) and with PIs (pSae2, 800 nM, **j**). Scale bar indicates 100 nm. **k** Representative atomic force microscopic image of pSae2 (50 nM). Examples of tetrameric proteins are circled. Scale bar indicates 100 nm. **l** Frequency distribution of pSae2 particle volumes based on experiments such as shown in **k**. Mean volume was calculated from nonlinear Gaussian fit (black curve). Error bars, SEM; 1543 particles were analyzed

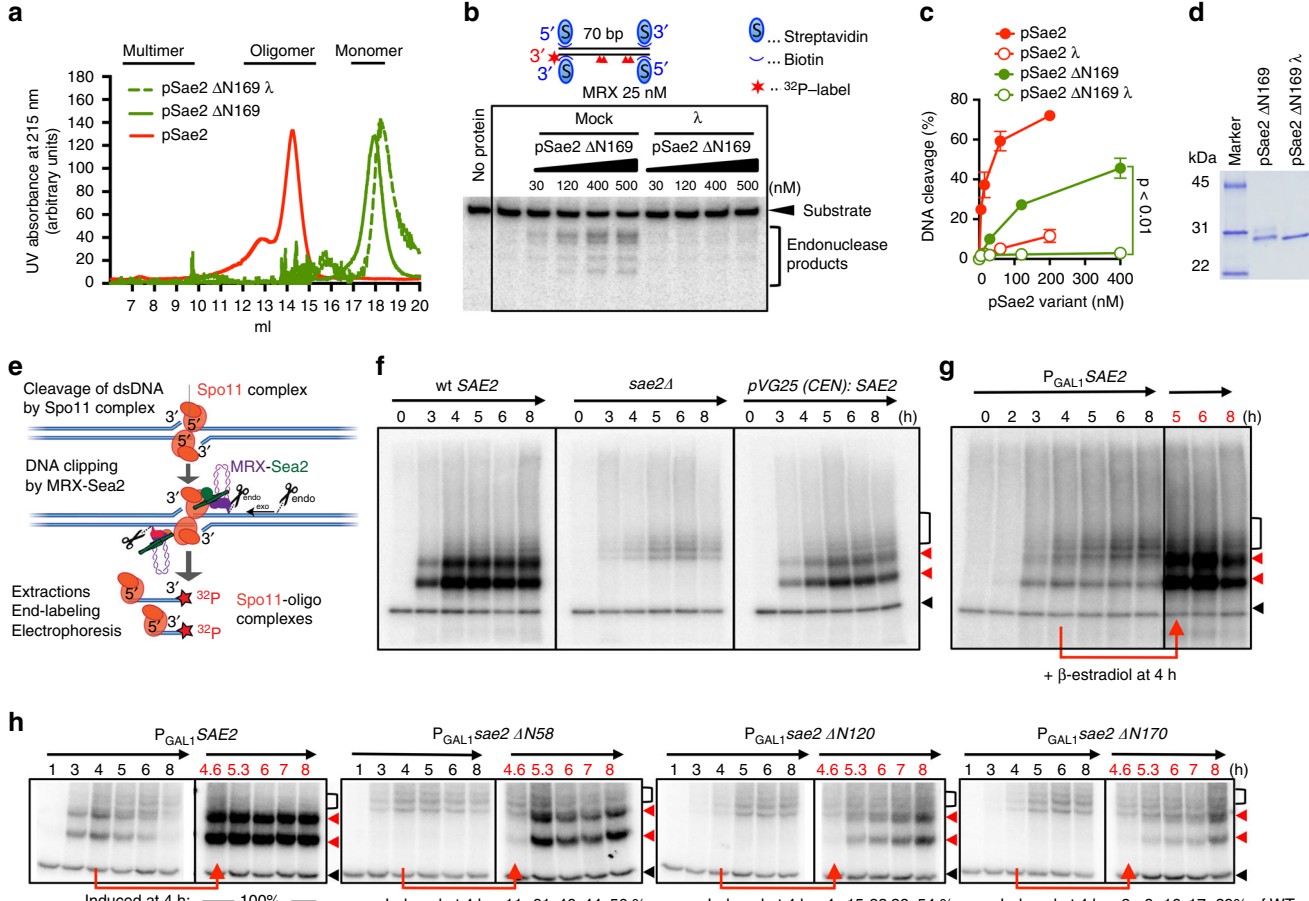

**Fig. 2** Sae2 oligomerization is required to promote the MRX nuclease in vitro and in vivo. **a** Size exclusion chromatography profile of the C-terminal domain of pSae2 (pSae2 ΔN169, residues 170–345) prepared with phosphatase inhibitors and either mock-treated (pSae2 ΔN169, green) or dephosphorylated with λ phosphatase (pSae2 ΔN169 λ, green dashed). The profile of full-length pSae2 (same as in Fig. 1h) is shown for reference. **b** Representative nuclease assays with pSae2 ΔN169 treated or not with λ phosphatase and MRX. **c** Quantitation of assays such as shown in **b**; error bars, SEM, $n \geq 4$. **d** Coomassie-stained polyacrylamide gel showing pSae2 ΔN169 treated or not with λ phosphatase. **e** A scheme of the Spo11-oligonucleotide assay. See text for details. **f** Spo11-oligonucleotide assay performed with wt SAE2 (left), sae2Δ (middle), or wt SAE2 expressed from a centromeric plasmid under the control of its natural promoter and terminator (right). The red triangles mark the long and short Spo11-oligonucleotide species generated in wild-type cells. The open bracket marks MRX-Sae2-independent "double-cut" Spo11-oligonucleotide species observed in wild type and sae2Δ (D.J., V.G., and M.J.N, manuscript in preparation). The black triangle marks a non-specific terminal deoxynucleotidyl transferase band. **g–h** Spo11-oligonucleotide assay as in **f**, performed in sae2Δ cells with wt SAE2 or N-terminal truncations expressed from an inducible promoter 4 h after the onset of meiosis. Samples from uninduced cells are labeled in black; samples from cells induced with β-estradiol are labeled in red. The numbers below represent the amount of Spo11-oligonucleotides relative to wild type

analysis. The deletion of 169 residues from the N-terminus of pSae2 abolished self-interaction and the pSae2 ΔN169 in gel filtration experiments migrated as a very small unit likely corresponding to a monomer, irrespectively of its phosphorylation status (pSae2 ΔN169 and pSae2 ΔN169 λ, Fig. 2a). This stands in contrast to pSae2 ΔC95, lacking 95 C-terminal residues, which eluted as very large multimeric species with a broad distribution in size exclusion chromatography (Supplementary Fig. 2d).

The phosphorylated pSae2 ΔN169 fragment was ~3-fold less capable to promote the endonuclease of MRX compared to wild-type (wt) pSae2 (Fig. 2b, c), although still showing a phosphorylation-dependent electrophoretic mobility shift (Fig. 2d). Importantly, the residual stimulatory capacity of the monomeric pSae2 ΔN169 fragment was still dependent on its phosphorylation, as its treatment with λ phosphatase resulted in an almost complete loss of activity (Fig. 2b, c). Similar results were obtained with the pSae2 L25P E171G mutant, corresponding to the *sae2–58* allele deficient in self-interaction and impaired cellular functions[30]. This pSae2 L25P E171G variant appeared much smaller than wild type in gel filtration and had a decreased capacity to promote MRX, but its residual activity was also still dependent on its phosphorylation (Supplementary Fig. 2e–i). Therefore, phosphorylation controls Sae2 function in resection on at least two levels. First, it regulates higher-order pSae2 oligomerization and the redistribution to active pSae2 oligomers from multimeric forms (>4N). Second, additionally and independently, phosphorylation regulates the capacity of pSae2 to promote MRX by another mechanism, which will be revealed below. This is evident from experiments showing that phosphorylation also promotes the resection capacity of monomeric pSae2 variants.

**Sae2 truncations promote resection of meiotic DSBs.** To define the function of Sae2 in the resection of blocked DNA ends in vivo, we used the Spo11-oligonucleotide formation assay[14,34]. Most phenotypes associated with *sae2* defects in vivo reflect its pleiotropic functions, including indirect regulation of checkpoint signaling associated with altered persistence of MRX and Rad9 bound to DNA ends[35–37]. Instead, the Spo11-oligonucleotide assay is a direct readout of the resection capacity of MRX-Sae2 in vivo. The assay scores the release of covalent Spo11-DNA fragments from meiotic DSB ends (Fig. 2e). It has been established previously that this requires both MRX and Sae2[14,38]. MRX-Sae2 cleave DNA adjacent to the Spo11 complex near DSBs or at locations further away[38]. In the latter case, the initial endonucleolytic cleavage is followed by exonucleolytic degradation by the 3'→5' MRX exonuclease toward the DNA end[38]. Cells from various stages of the meiotic prophase were lysed, and FLAG-tagged Spo11-oligonucleotide complexes were enriched by immunoprecipitation. The ssDNA component was labeled at the 3' end by terminal deoxynucleotide transferase and [α-$^{32}$P]dCTP (Fig. 2e)[34]. As established previously, two major *SAE2*-dependent products can be observed[14]. The deficiency in the Spo11-oligonucleotide formation in *sae2Δ* mutants can be complemented by *SAE2* expressed from a plasmid (Fig. 2f, g). We used either a centromeric vector (pVG25 CEN:*SAE2*), where *SAE2* expression was driven by its cognate promoter (Fig. 2f), or from a genomic β-estradiol-inducible promoter (P$_{GAL1}$:*SAE2*) that leads to higher levels of *SAE2* expression and generally results in a better efficiency of complementation (Fig. 2g). In accord with our reconstituted reactions, various truncations of the N-terminal domain of Sae2 resulted in a decrease but not a complete elimination of the Spo11-oligonucleotide signal in experiments where the Sae2 truncations were overexpressed (Fig. 2h). The self-interaction impaired Sae2 L25P mutation resulted in a complete deficiency in resection of Spo11-bound DSBs, even when the Sae2 mutant was overexpressed (Supplementary Fig. 3a, b). This showed that the L25P mutation also in vivo brings about end-processing defect that is comparable to a complete elimination of the entire 170 residues long N-terminal domain. These results collectively suggest that proper Sae2 oligomerization is required for its optimal capacity to promote resection of meiotic DSBs in conjunction with MRX in vivo.

**Sae2 phosphorylation by CDK is not sufficient for resection.** CDK-dependent phosphorylation of Sae2 at the CDK consensus site of S267 is known to be critical for its function in DNA end resection and homologous recombination[6], but the mechanism of how phosphorylation regulates Sae2 function has not been defined. In accord with previous work[6,24], we show that the non-phosphorylatable alanine substitution of S267 dramatically reduced the capacity of pSae2 to promote the MRX endonuclease (Fig. 3a, b). This stands in contrast to the other two CDK consensus sites in pSae2, S134, and S179, which had no effect on pSae2 activity when mutated to alanine (Fig. 3a, b). In contrast, the phosphomimetic glutamic acid substitution mutant (pSae2 S267E) supported resection, and its activity was partially resistant to treatment with λ phosphatase (Fig. 3c–e)[6]. The ~5-fold decrease in MRX-dependent DNA cleavage upon dephosphorylation of pSae2 S267E demonstrated that other sites in pSae2, in addition to S267, must be phosphorylated for its optimal stimulation of the MRX endonuclease. Accordingly, the activity of the phosphomimetic double mutants pSae2 S134E S267E and pSae2 S179E S267E mutants were ~1.5-fold more resistant to λ phosphatase treatment than pSae2 S267E alone (Supplementary Fig. 4a–c), suggesting that the other two putative CDK sites may play a supporting function. However, the capacity of all phosphomimetic CDK site variants to stimulate MRX decreased upon λ phosphatase treatment, showing that other non-CDK sites must also be involved.

Furthermore, we show that pSae2 that had been dephosphorylated by λ phosphatase can be subsequently partially activated upon phosphorylation by human recombinant CDK1/Cyclin B in vitro (Fig. 3f, g). This suggests that the size distribution transitions of pSae2 are likely dynamic, and active pSae2 may be obtained not only by de novo protein synthesis but also upon phosphorylation of the soluble multimer pool of Sae2[8].

In accord with the observations in vitro, the *sae2 S267A* cells were deficient in the Spo11-oligonucleotide formation assay, even when Sae2 S267A was overexpressed (Fig. 3h, Supplementary Fig. 4d, e). In contrast, *sae2 S267E* cells supported resection of meiotic Spo11-bound DSBs, albeit to a lesser degree than wt (Fig. 3h, Supplementary Fig. 4d). Resection of meiotic DSBs was not largely affected by substitutions in S134 and S179 of pSae2 (Fig. 3h), similar to the in vitro experiments (Fig. 3a, b).

To define how phosphorylation of pSae2 at S267 promotes its function in resection, we analyzed the size distribution of recombinant pSae2 S267E and pSae2 S267A. Both pSae2 variants showed a comparable phosphorylation-dependent shift in electrophoretic mobility in polyacrylamide gels (Fig. 3e) and eluted in the oligomeric fraction during gel filtration just like wt pSae2 (Supplementary Fig. 4f, g). Furthermore, the S267E mutation did not prevent the pSae2 variant from multimerizing upon dephosphorylation with λ phosphatase (Supplementary Fig. 4g). This suggested that phosphorylation of pSae2 at the CDK site of S267 regulates pSae2 by a mechanism that is distinct from any effects on oligomerization. To confirm this, we next analyzed the effect of the S267 substitution mutations in the monomeric C-terminal fragment of pSae2. We observed that the S267A substitution largely eliminated the residual capacity of the pSae2 ΔN169 to promote MRX endonuclease activity on protein-blocked ends (Supplementary Fig. 4h, i), while the S267E mutation had a smaller effect (Supplementary Fig. 4j, k). We note that the S267E substitution had no apparent effect on the function of the full-length pSae2 polypeptide (Fig. 3c, d), but reduced the resection capacity of the C-terminal fragment in conjunction with MRX at least three-fold (Supplementary Fig. 4j, k), reminiscent of the reduction of the resection capacity caused by the S267E mutation in vivo (Fig. 3h). Notably, λ phosphatase treatment of the pSae2 ΔN169 S267E mutant marginally reduced

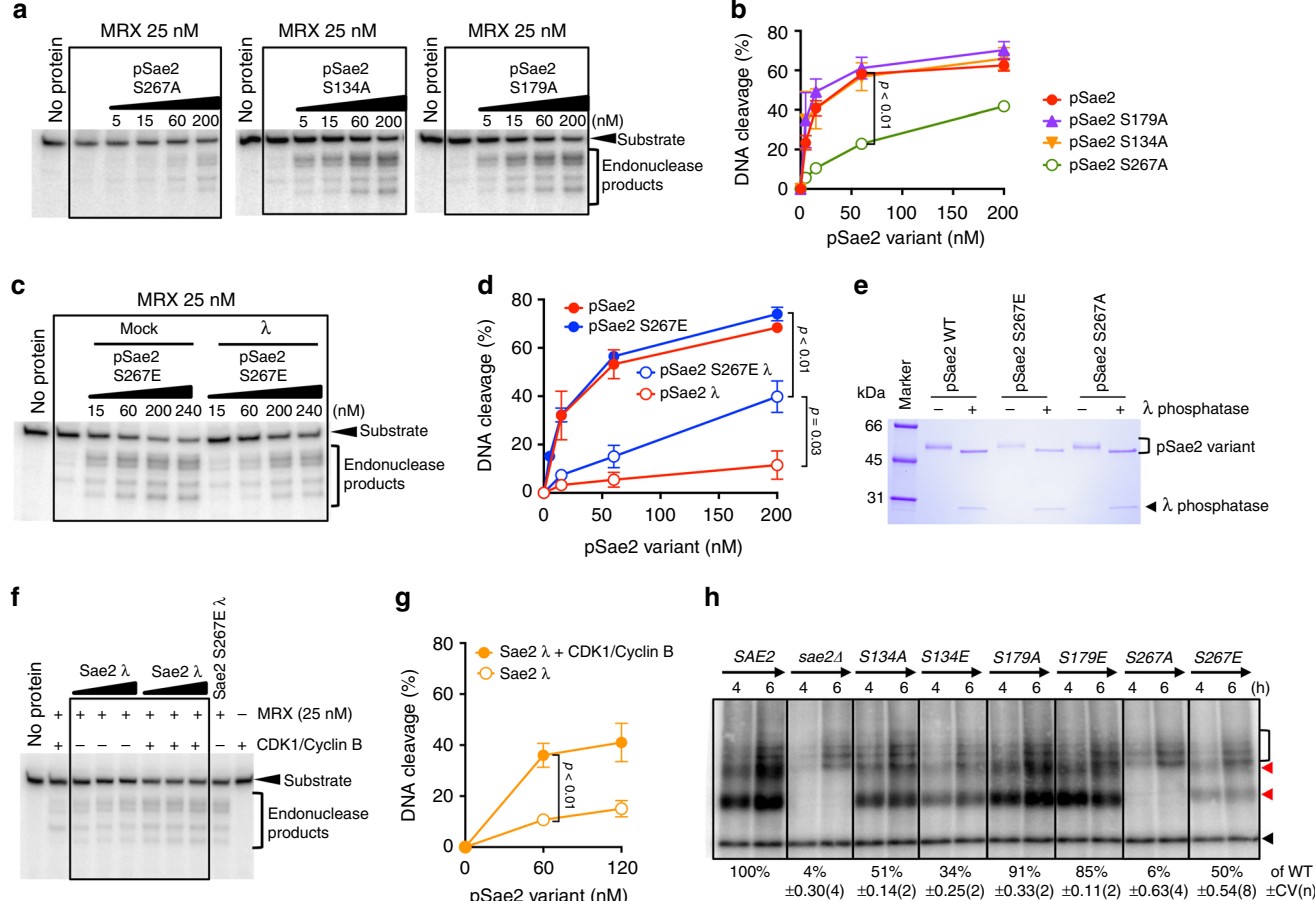

**Fig. 3** CDK phosphorylation of Sae2 is prerequisite, but not sufficient, to promote the MRX nuclease. **a** Nuclease assays with MRX and pSae2 variants mutated in one of the three CDK consensus sites (S267, S134, and S179) into alanine. All mutants were expressed in the presence of phosphatase inhibitors. Substrate used was the same as in Fig. 1c. **b** Quantitation of the assays such as shown in **a**; error bars, SEM; $n \geq 3$. **c** Nuclease assays with pSae2 S267E phosphomimicking variant, expressed in the presence of phosphatase inhibitors. The pSae2 S267E protein was mock or λ phosphatase treated, as indicated. **d** Quantitation of assays such as shown in **c**; error bars, SEM; $n \geq 3$. **e** Representative polyacrylamide gel (4–15%) electrophoretic analysis of the indicated pSae2 variants, treated or not with λ phosphatase, stained with Coomassie brilliant blue. **f** Sae2 was expressed and dephosphorylated with λ phosphatase during purification (Sae2 λ). The polypeptide was then mock-treated or phosphorylated with recombinant CDK1/Cyclin B, where indicated, and used in nuclease assays (60, 120, and 200 nM, respectively) with MRX. The pSae2 S267E variant treated with λ phosphatase (pSae2 S267E λ, 200 nM) was used as a reference. **g** Quantitation of assays such as shown in **f**; error bars, SEM; $n \geq 3$. **h** Spo11-oligonucleotide assay, as in Fig. 2f, performed in sae2Δ cells with wt, and sae2 mutant bearing non-phosphorylatable (S→A) or phosphomimicking (S→E) mutations in the CDK consensus sites of Sae2 expressed from a centromeric vector. Quantitation is shown below the lanes as an average (relative to wild type) ± coefficient of variation

its function to promote MRX; this decrease, however, remained statistically significant (Supplementary Fig. 4j, k). Together, these results demonstrate that CDK-dependent phosphorylation of Sae2 at S267 is critical but not sufficient to maximally promote DNA end resection by MRX and that this occurs by a mechanism that is distinct from the regulation of Sae2 oligomerization.

**Sae2 phosphorylation by Mec1/Tel1 has little role in resection.** Single or combined mutations in the putative Mec1 or Tel1 sites at SQ/TQ motifs at positions S73, T90, S249, and T279 of pSae2 did not affect its capacity to regulate the MRX nuclease in vitro (Fig. 4a–f, Supplementary Fig. 5a). Likewise, the quintuple pSae2 S73E T90E S249E T279E S267E mutant was no more active than pSae2 S267E upon dephosphorylation with λ phosphatase (Fig. 4c–e, see also Supplementary Fig. 5b–e), although the four phosphomimetic amino acid substitutions slightly prevented pSae2 multimerization upon dephosphorylation (Supplementary Fig. 5f, g). As above, the effects of the sae2 mutations were more pronounced in vivo. Single non-phosphorylatable substitutions of S73A and T279A brought about defects in Spo11 release that were

more severe than respective phosphomimetic mutations, while T90 was sensitive to both substitutions and S249 was not important (Fig. 4g, h). However, the non-phosphorylatable T279A substitution (together with S278A) was able to complement sae2Δ when overexpressed, indicating that phosphorylation of these sites is not absolutely essential to regulate Sae2 (Fig. 4i) but likely plays a supporting function. Nevertheless, both Mec1- and Tel1-dependent modifications promote the release of Spo11-oligonucleotides, as tel1Δ or mec1 cells show a reduction in the Spo11-oligonucleotide signal, while simultaneous MEC1 and TEL1 deficiency almost completely eliminated meiotic DSB processing (Supplementary Fig. 5h), in agreement with previous results[39].

While most phosphorylation sites in pSae2 promote its function in resection, one notable exception is the SQ site at S289. While the non-phosphorylatable substitution S289A had no effect in the Spo11-oligonucleotide assay, phosphomimicking sae2 S289D or S289E variants were completely defective (Fig. 4j). The same effects were observed with the respective recombinant protein variants in vitro in conjunction with MRX (Fig. 4k, l). The S289E mutation was inhibitory also in combination with the

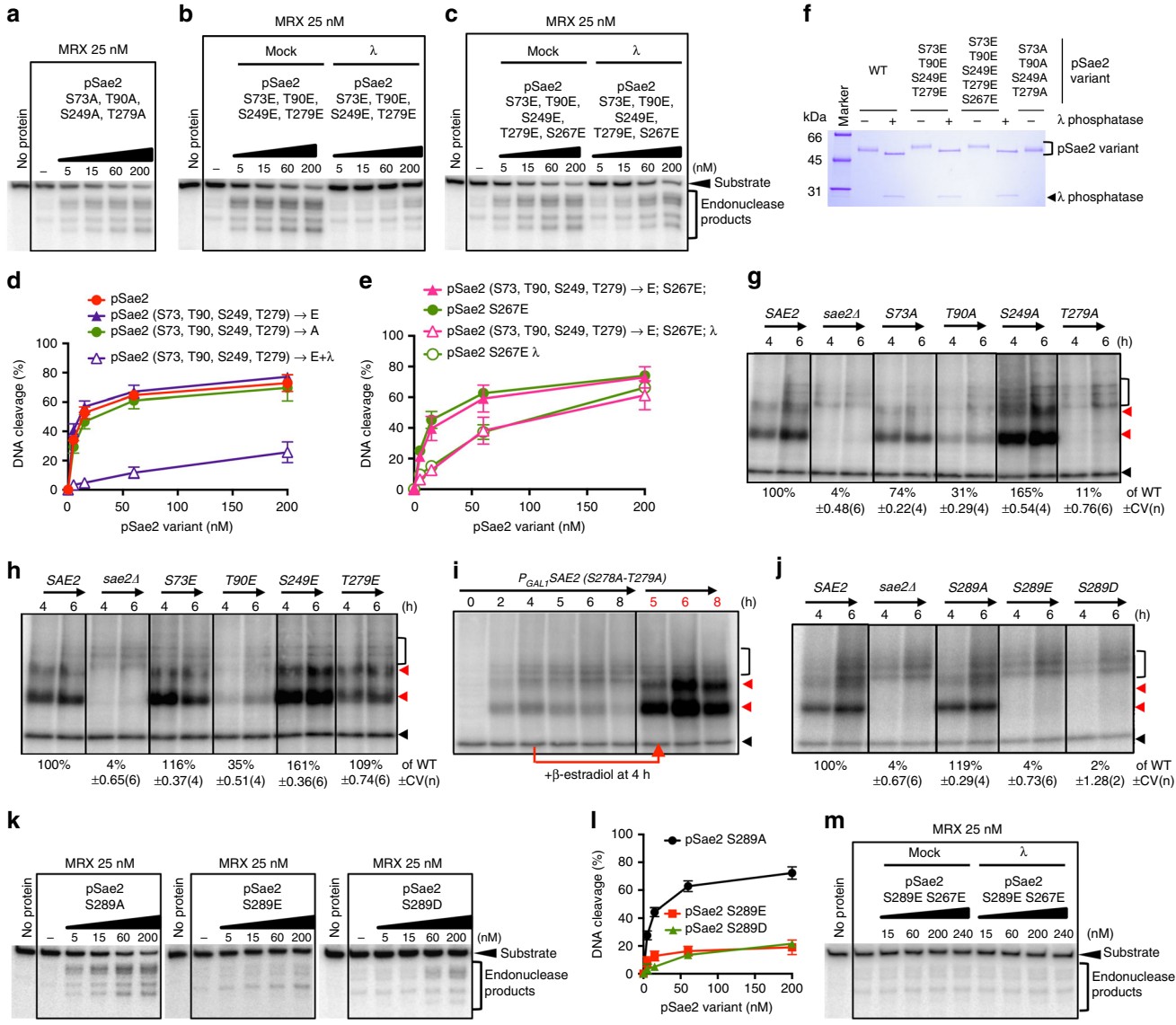

**Fig. 4** Mec1/Tel1 phosphorylation of Sae2 stimulates, but is not essential, for its capacity to promote the MRX nuclease. **a** Nuclease assays with MRX and pSae2 mutated simultaneously at SQ/TQ sites at positions 73, 90, 249, and 279, preventing phosphorylation due to alanine substitutions. The pSae2 mutant was prepared with phosphatase inhibitors. Substrate used was the same as in Fig. 1c. **b** Nuclease assays with MRX and pSae2 mutated simultaneously at SQ/TQ sites at positions 73, 90, 249, and 279 mimicking phosphorylation due to glutamic acid substitutions. The pSae2 mutant was prepared with phosphatase inhibitors and then treated or mock-treated with λ phosphatase. **c** Assays as in **b**, with a pSae2 variant carrying in addition a phosphomimicking mutation at the CDK site of S267. **d** Quantitation of assays such as shown in **a**, **b**; error bars, SEM; n ≥ 3. **e** Quantitation of assays such as shown in **c**; error bars, SEM; n ≥ 3. **f** Representative polyacrylamide gel (4–15%) electrophoretic analysis of the indicated pSae2 variants, treated or not with λ phosphatase, stained with Coomassie brilliant blue. **g–j** Spo11-oligonucleotide assay, as in Fig. 2f–h, with sae2Δ cells bearing a centromeric plasmid expressing **g–j** individual non-phosphorylatable (S→A) or **h–j** phosphomimicking mutations in the SQ/TQ sites of Sae2 at positions 73, 90, 249, 279, and 289, or **i** with sae2 S278A T279A variant expressed upon addition of β-estradiol 4 h after the onset of meiosis. Quantitation is shown below the lanes as an average (relative to wild type) ± coefficient of variation. **k** Nuclease assay with pSae2 S289A, pSae2 S289E, and pSae2 S289D mutants and MRX. All pSae2 variants were expressed and purified in the presence of phosphatase inhibitors. **l** Quantitation of assays such as shown in **k**; error bars, SEM; n ≥ 3. **m** Nuclease assay with pSae2 S289E S267E double mutant prepared in the presence of phosphatase inhibitors. The Sae2 variant was then mock-treated or dephosphorylated with λ phosphatase

phosphomimetic CDK site S267E mutant (Fig. 4m), and this modification might thus represent a negative regulatory signal to turn off the MRX-Sae2 nuclease. Alternatively, the E/D substitutions may cause protein misfolding, and it therefore remains to be established whether S289 phosphorylation occurs as a regulatory mechanism in cells.

In addition to promoting the Mre11 nuclease, Sae2 was reported to possess an intrinsic nuclease activity, and the Sae2 mutant (N123A R127A) abolished this function[40]. However, we found the

sae2 N123A R127A mutant behaved as wild type in the Spo11-oligonucleotide formation assay (Supplementary Fig. 5i). We therefore conclude that Sae2 functions in resection as an activator of Mre11, rather than having an intrinsic catalytic function[24].

**Multiple phosphorylations regulate Sae2 role in resection.** Dephosphorylation of all pSae2 phosphomimetic substitution mutants and their combinations analyzed to this point reduced

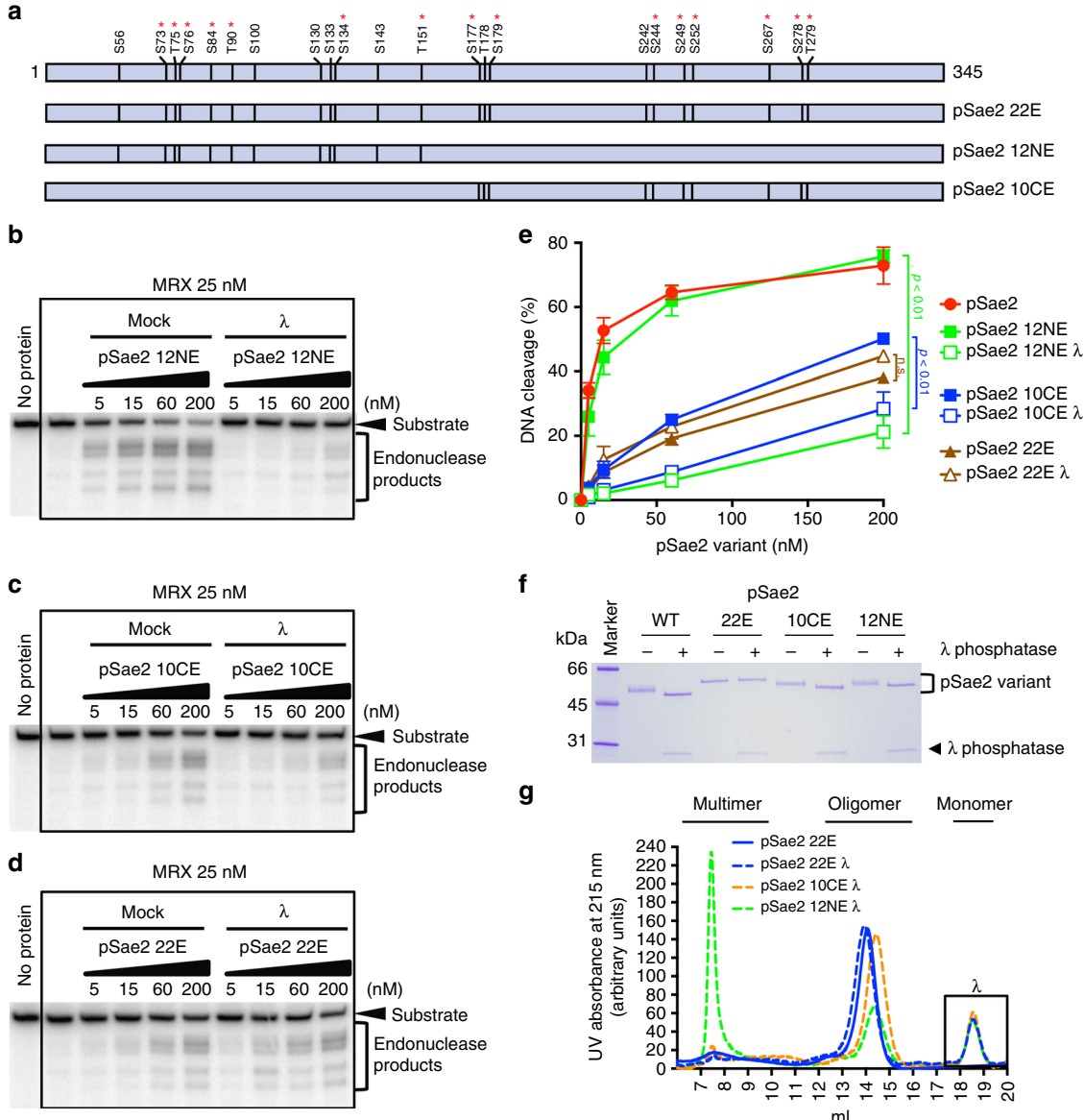

**Fig. 5** Multiple residues in Sae2 must be phosphorylated to prevent multimerization and promote MRX. **a** Overview of putative phosphorylation sites in pSae2 based on our mass spectrometric analysis and previous reports[8,12,41]. The red asterisk indicates pSae2 residues found phosphorylated in vivo in previous studies. The pSae2 22E mutant contains 22 phosphomimicking substitutions. The pSae2 12NE mutant contains 12 phosphomimicking substitutions in the N-terminal part of the protein. The pSae2 10CE mutant contains 10 phosphomimicking substitutions in the C-terminal part of the protein. All mutants were prepared in the presence of phosphatase inhibitors. **b**–**g** Analysis of the pSae2 22E, pSae2 12NE, and pSae2 10CE mutants. The polypeptides were mock-treated or dephosphorylated with λ phosphatase and subjected to nuclease assays with MRX (**b**–**d**). Substrate used was the same as in Fig. 1c. **e** shows quantitation of the experiments such as shown in **b**–**d**, error bars, SEM; $n = 3$. **f** shows a representative polyacrylamide gel (4–15% gradient) electrophoretic analysis of the respective mutants treated or not with λ phosphatase. **g** shows a representative size exclusion chromatography profile of the indicated mutants mock-treated or dephosphorylated with λ phosphatase. The rectangle indicates the position of λ phosphatase

the capacity of Sae2 to promote the MRX nuclease and resulted in increased electrophoretic mobility. This indicated that additional residues in our recombinant pSae2 are phosphorylated and facilitate pSae2 function to promote MRX. To define these phosphorylation sites, we subjected our recombinant protein to mass spectrometry and identified multiple sites that are likely to be modified (Supplementary Fig. 6a, b). Notably, many of these sites overlapped with positions identified in previous studies[8,12,41]. However, as our protein was expressed in insect cells, we cannot exclude that some of the non-overlapping sites are not modified in yeast or are rapidly dephosphorylated in cells.

We observed that seven non-phosphorylatable alanine substitutions in the pSae2 N-terminus had no effect on its activity,

while six alanine substitutions in the C-terminal region of the full-length polypeptide completely eliminated its stimulatory effect on MRX (Supplementary Fig. 6c–e). This showed that phosphorylation of pSae2 at its C-terminal rather than the N-terminal part is more important for its function in resection.

For subsequent analysis, we selected 22 putative pSae2 phosphorylation sites based on our mass spectrometric analysis and previous work; 15 of those pSae2 sites had been previously found to be modified (Fig. 5a)[8,12,41]. Thirteen of these sites were identified in Flag-Sae2 pulldown from yeast cells[8] and thus very likely to be modified in vivo. We next expressed and purified three full-length phosphomimetic pSae2 variants in the presence of phosphatase inhibitors, with the respective substitution

mutations in the N- or C-terminal parts of the full-length polypeptide (pSae2 12NE or pSae2 10CE, respectively) or with all 22 mutations combined (pSae2 22E) (Fig. 5a). We observed that the 12 N-terminal pSae2 substitutions failed to render the pSae2 variant activity resistant to dephosphorylation (Fig. 5b–f). Furthermore, the λ phosphatase-treated pSae2 12NE multimerizes, as revealed by size exclusion chromatography (Fig. 5g, Supplementary Fig. 6f). In contrast, the phosphomimetic pSae2 10CE mutant retained partial activity upon dephosphorylation and did not multimerize (Fig. 5c, e–g, Supplementary Fig. 6f), underscoring the importance of C-terminal pSae2 phosphorylation in regulating activity and dephosphorylation. Nevertheless, even for the pSae2 10CE mutant, a reduced capacity to stimulate MRX following 10CE dephosphorylation indicated that additional phosphorylation events promote maximal activity (Fig. 5c–e). Consistent with this hypothesis, λ phosphatase treatment shifted pSae2 10CE in a polyacrylamide gel (Fig. 5f).

Finally, we analyzed the pSae2 22E variant, which possessed all 22 phosphomimetic substitution mutations. Strikingly, the pSae2 22E variant did not change its mobility in a polyacrylamide gel upon λ phosphatase treatment (Fig. 5f). The pSae2 22E mutant was still able to promote MRX, although ~10-fold less efficiently that wt pSae2, but the capacity of the pSae2 22E variant to stimulate MRX was not sensitive to λ phosphatase (Fig. 5d, e). Also, the pSae2 22E protein did not multimerize upon λ phosphatase treatment (Fig. 5g, Supplementary Fig. 6f), showing that our analysis identified all residues that were important for pSae2 function. We conclude that a number of residues both in the N- and C-terminal parts of pSae2 must be phosphorylated for its maximal stimulatory effect on the MRX nuclease. Phosphorylation sites in the C-terminal part of pSae2 are more important than those in the N-terminal region.

**Phosphorylated C-terminus of Sae2 interacts with Rad50.** To define oligomerization-independent functions of Sae2 phosphorylation, we set out to test whether phosphorylation regulates a direct physical interaction between Sae2/pSae2 and the MRX complex. The interaction between MRX and Sae2 is rather weak. Using recombinant proteins, we previously found that Sae2 could interact with the Mre11 and Xrs2 subunits but failed to detect an interaction with Rad50[24]. To determine whether hyperphosphorylation of Sae2 affects its interaction with MRX, we immobilized recombinant phosphorylated or non-phosphorylated maltose-binding protein (MBP)-tagged Sae2 on amylose resin and tested for their interaction with the MRX complex. As shown in Fig. 6a, the MRX complex interacted with Sae2 irrespectively of Sae2 phosphorylation.

As MRX and Sae2 interact via multiple interfaces[24], we next set out to test the interaction between the C-terminal pSae2 domain (pSae2 ΔN169), i.e., the minimal unit that is required to promote MRX activity, with individual subunits of the MRX complex. Xrs2 is not essential for the MRX-Sae2 endonuclease[42], so we tested for interactions with Mre11 and Rad50. Unlike full-length Sae2[24], we found that the C-terminal fragment did not interact with Mre11 (Supplementary Fig. 7). To test for interaction with Rad50, we expressed and purified FLAG-tagged Rad50 and immobilized it on anti-FLAG affinity resin. The resin was then incubated with phosphorylated (mock-treated) or λ phosphatase-treated pSae2 ΔN169 (Fig. 6b). Strikingly, we observed that the C-terminus of Sae2 could interact with Rad50 only in its phosphorylated state. The same results were obtained when the experiment was performed reciprocally, i.e., when his-tagged pSae2 ΔN169 was immobilized on NiNTA resin and then incubated with Rad50. Also in this case, the C-terminus of Sae2 interacted with Rad50 only in its phosphorylated state (Fig. 6c). Importantly, the

phosphorylation-dependent interaction between the pSae2 ΔN169 and Rad50 was not affected by the Rad50 K40A mutation, which prevents ATP binding and renders the MRX nuclease refractory to stimulation by pSae2 (Fig. 6c)[24].

In 1990, Kleckner and colleagues isolated a group of rad50 separation of function mutants (rad50S) that blocked spore formation in meiosis but exhibited less severe sensitivity to the methylating agent methyl methane sulfonate compared to a rad50Δ strain[15]. Later, it was shown that the rad50S cells are impaired in the processing of DNA ends blocked by proteins including topoisomerases and Spo11[14,16,20–22], and the corresponding M-Rad50S-X variant (containing Rad50 K81I mutant, representative of Rad50S) was deficient in DNA clipping in conjunction with Sae2 in vitro[24]. The Rad50 K81I mutation lies in a surface patch, and it is not expected to be deficient in ATP hydrolysis but rather was proposed to affect a protein–protein interaction[43]. In fact, Sae2 overexpression could partially suppress single-strand annealing defects of rad50S cells, suggesting that the MRX-Sae2 interaction may be affected[39]. Strikingly, we observed that recombinant Rad50S was impaired in its interaction with the C-terminal part of Sae2, even when it was phosphorylated (Fig. 6d). Therefore, we conclude that the Rad50S mutant is deficient in the physical interaction with phosphorylated Sae2, which explains why the M-Rad50S-X complex is refractory to regulation by pSae2[24], and therefore defective in meiosis[14,16]. Together, our results demonstrate that, while phosphorylation of Sae2 does not affect its overall capacity to interact with MRX, it controls the specific physical and functional interaction with the MRX subunit Rad50, which is a prerequisite for the regulation of the Mre11 nuclease.

## Discussion

DNA end resection commits the repair of broken DNA to the recombination pathway[3]. To prevent unscheduled DNA degradation and aberrant recombination, the initiation of DNA end resection must be under a tight control. It has been established that phosphorylation of Sae2/CtIP is a key component of this control mechanism[6]. However, how phosphorylation promotes Sae2 function to stimulate DNA end resection has remained elusive. We show that phosphorylation reduces the capacity of pSae2 to bind DNA, which may be due to conformational change or a negative overall charge of the hyperphosphorylated polypeptide. We also cannot exclude that non-phosphorylated or weakly phosphorylated Sae2 aggregates on DNA, which increases its apparent DNA-binding activity. However, as DNA cleavage positions are solely determined by the MRX complex and Sae2 only promotes cleavage efficacy, we favor the hypothesis that DNA binding by Sae2, at least in the simple reconstituted system, is not required for the clipping function of MRX.

Here we report that extensive phosphorylation of Sae2 controls its capacity to promote the MRX nuclease by at least two distinct mechanisms (Fig. 7). During the G1 phase of the cell cycle, Sae2 exists in an unphosphorylated state and is part of an inactive soluble multimeric complex[8]. During S phase in mitotic cells and the prophase of the first meiotic division, as well as upon DNA damage[6,10], Sae2 is extensively phosphorylated resulting in an electrophoretic mobility shift[8,11]. First, we show that phosphorylated pSae2 transitions toward smaller molecular weight complexes. Our AFM analysis shows that phosphorylation promotes the formation of pSae2 tetramers in solution without DNA, which likely represent the active pSae2 species that optimally promote the Mre11 nuclease within the MRX complex. MRX then initiates DNA end resection, which is especially important for the processing of protein-bound DNA ends. Our results are in agreement with previous observations that Sae2 oligomerization

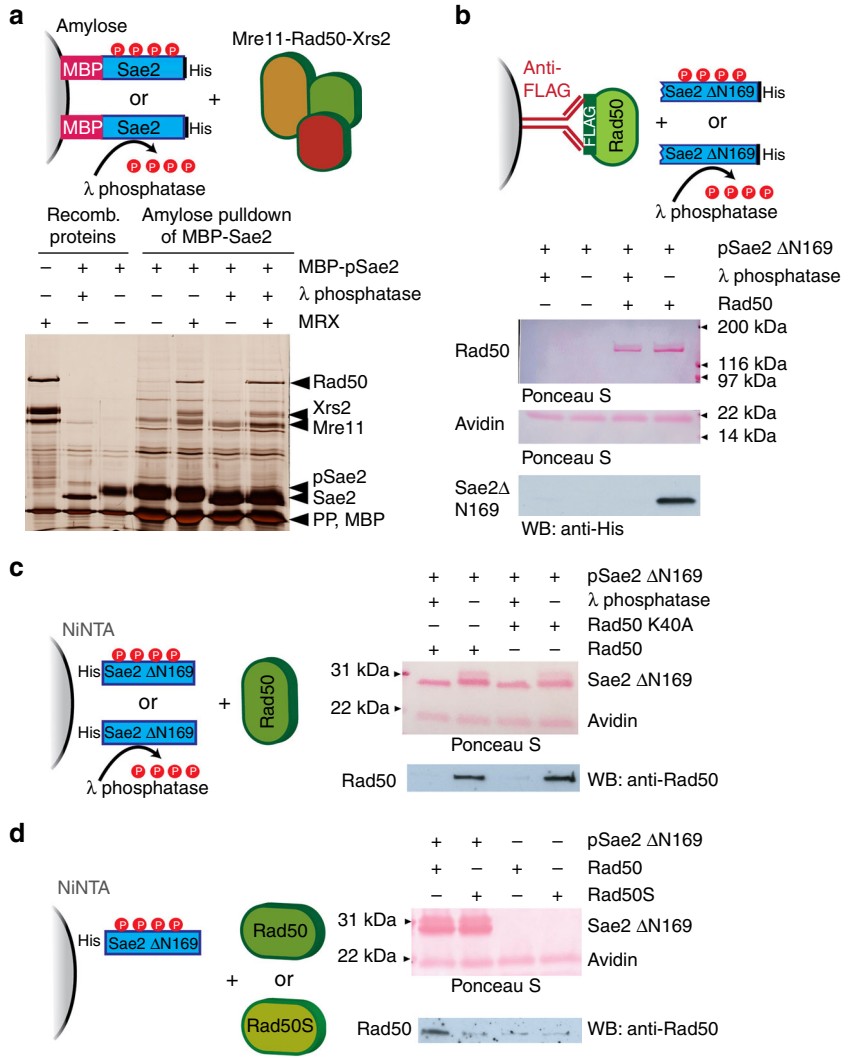

**Fig. 6** Phosphorylated C-terminus of Sae2 interacts with Rad50. **a** Full-length phosphorylated recombinant MBP-tagged pSae2 was mock-treated or dephosphorylated with λ phosphatase upon binding to amylose resin and incubated with recombinant MRX complex. The eluates were visualized by silver staining. Prescission protease was added to all samples as a protein stabilizer and to cleave the MBP tag off pSae2. **b** The FLAG-tagged recombinant Rad50 protein was immobilized on anti-FLAG affinity resin and incubated with phosphorylated C-terminal domain of pSae2 (pSae2 ΔN169, residues 170–345), which had been either mock-treated or dephosphorylated with λ phosphatase. The bound proteins were eluted and detected by Ponceau staining or western blotting. Avidin was added to elution buffer and shows equal loading. **c** The phosphorylated recombinant MBP-tagged C-terminal domain of pSae2 (residues 170–345) was bound to amylose resin, eluted, cleaved with prescission protease, and immobilized on NiNTA resin. The bound pSae2 ΔN169 was mock-treated or dephosphorylated with λ phosphatase and incubated with recombinant wild-type Rad50 or ATP binding-deficient Rad50 K40A. Proteins were eluted and visualized by Ponceau staining or western blotting. Avidin was added to elution buffer and shows equal loading. **d** Assay as in **c**. Phosphorylated C-terminal domain of pSae2 was incubated with wild-type Rad50 or Rad50 K81I (representative Rad50S) mutant

is critical for its activity[30] and observations with Sae2 orthologs in *S. pombe* and humans[32,33].

Second, we also show that additionally and independently of regulating size distribution, phosphorylation of the C-terminal region of Sae2 is necessary for a direct physical interaction with the Rad50 subunit of MRX. Sae2 interacts with all components of the MRX complex including Mre11 and Xrs2[24], but the phosphorylation-dependent interaction with Rad50 is particularly important for DNA end resection. ATP hydrolysis by Rad50 is necessary for the MRX-Sae2 endonuclease[24]. Because phosphorylated Sae2 also promotes the exonuclease of Mre11 when ATP hydrolysis is allowed (Cannavo et al., manuscript in preparation), we suggest that phosphorylated Sae2 controls the Mre11 nuclease by promoting productive ATP hydrolysis by Rad50. The direct physical interaction between Sae2 and Rad50 thus likely mediates this process—an idea that is supported by our

observation that the phosphorylation-dependent interaction between Sae2 and Rad50 is abolished by the *rad50S* mutation. These observations also explain why the M-Rad50S-X complex fails to process meiotic Spo11-bound DSBs in vivo[14,16] and is refractory to stimulation by phosphorylated Sae2 in vitro[24].

We define the function of the individual phosphorylation sites in Sae2 found by mass spectrometric analysis and in previous studies. Although we found 22 sites that are likely to be modified in the population of recombinant pSae2, most individual polypeptides will contain only a fraction of those modifications (Supplementary Fig. 6b). We observed that, while Sae2 mutations had a more pronounced effect in the Spo11-oligonucleotide assay in vivo than in the reconstituted biochemical assays, there was generally a good correlation between both approaches. Phosphorylation of the conserved CDK site of Sae2 at S267 is critical for its resection function in vitro and in vivo[6], but there is a need

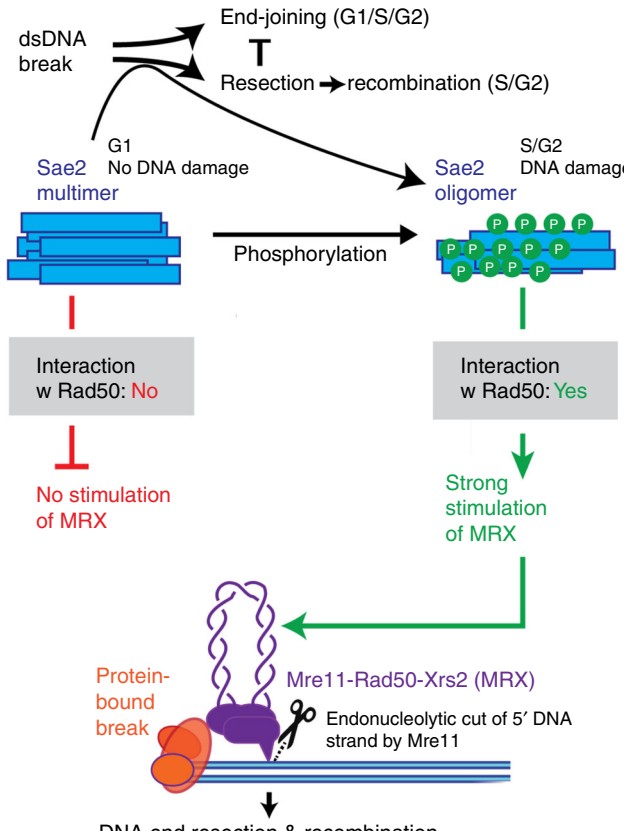

**Fig. 7** The function of Sae2 phosphorylation in the control of DNA end resection. Phosphorylation allows the formation of a tetrameric Sae2 form, which is capable of promoting the nuclease activity of Mre11-Rad50-Xrs2 (MRX). Additionally, phosphorylation of the C-terminal part of Sae2 is prerequisite for interaction with Rad50, which is also essential to promote MRX. These phosphorylation-dependent transitions mediate cell cycle and DNA-damage-dependent control of DNA end resection capacity of MRX. This prevents unscheduled DNA degradation by Mre11, aberrant recombination and resulting genome instability

for the modification of a number of additional sites that play a supporting role. This may include other CDK sites, Mec1/Tel1 sites, or other kinases such as Cdc5 or the Dbf4-depenent kinase, whose function in Sae2 regulation has, however, not yet been defined. Although phosphorylation of sites in the C-terminal part of Sae2 is more important, multiple residues both in the N and C-terminal domains of Sae2 must be modified for optimal activity. We propose that these additional sites need to alter the charge of the Sae2 polypeptide, which then affects its size distribution, rather than playing a sequence-specific function.

Collectively, our results support the view that the primary function of Sae2 in DSB resection and meiosis is mediated by the stimulation of the Mre11 nuclease[24] and not an intrinsic catalytic function of Sae2[40]. In accord, the phosphorylation-dependent regulatory control of DNA end resection utilizes Sae2 as an activator of MRX. In the absence of a pro-resection phosphorylation signal, the Mre11 nuclease activity is restricted. Once activated upon Sae2 phosphorylation, Mre11 initiates resection, which commits DSB repair to the recombination pathway (Fig. 7). Our results will be relevant for understanding the regulation of DNA end resection in human cells. Whereas the Xrs2 subunit of MRX is dispensable for resection in *S. cerevisiae* as long as nuclear import of Mre11-Rad50 is guaranteed[42], NBS1 is in contrast essential for resection in humans[28,29]. This likely reflects more complex control mechanisms in high eukaryotes;

however, basic elements of the resection process are well conserved between yeast and humans. The analysis reported here will thus be invaluable to understand how phosphorylation controls the interplay of MRN and CtIP, defects in which are linked to multiple human pathologies.

## Methods

**Preparation of recombinant proteins**. The *S. cerevisiae* Mre11-Rad50-Xrs2 complex contained a His-tag and a FLAG-tag at the C-termini of Mre11 and Xrs2, respectively. MRX was expressed in *Sf*9 cells and purified using affinity chromatography[44]. Recombinant pSae2, containing a MBP tag at the N-terminus and a His-tag at C-terminus, was expressed in *Sf*9 cells and purified by sequential affinity chromatography steps using amylose resin (New England Biolabs), followed by the cleavage of the MBP tag with PreScission protease (12 µg of protease per 100 µg MBP–Sae2–His). Finally, pSae2 was purified using NiNTA agarose (Qiagen). We modified our original procedure[24] to preserve maximal phosphorylation of the protein. Specifically, Okadaic acid (100 nM, Calbiochem) was added to the *Sf*9 cells for the last 3 h of culture before harvesting. The lysis buffer was supplemented with the following phosphatase inhibitors: okadaic acid (100 nM), sodium orthovanadate (1 mM, Sigma), sodium fluoride (20 mM, Sigma), and sodium pyrophosphate (15 mM, Applichem). Where indicated, pSae2 was dephosphorylated with λ phosphatase (New England Biolabs). To this point, 1.3 µg of pSae2 was incubated at 30 °C for 15 min with 280 U of λ phosphatase (New England Biolabs) in the manufacturer's recommended buffer in a volume of 20 µl (final pSae2 concentration 1.6 µM). As a control, mock-treated pSae2, incubated in the λ phosphatase reaction buffer, but without λ phosphatase, was used. The Sae2 variants were then returned on ice and immediately added to the nuclease reaction mixture. We note that pSae2 did not lose its capacity to promote MRX during mock treatments (compare Fig. 1d with Supplementary Fig. 2g). The Sae2 variant that has been obtained upon dephosphorylation of the originally phosphorylated pSae2 is denoted as "pSae2 λ". Alternatively, to obtain non-phosphorylated Sae2 already during protein purification, phosphatase inhibitors were omitted from the cell culture medium and the lysis buffer. In addition, manganese chloride (1 mM) and λ phosphatase (New England Biolabs, 20,000 units for purification from 1.6 l *Sf*9 cell culture) were added to MBP-Sae2 after elution from amylose resin, before the addition of Prescission protease. Incubation was carried out first at 30 °C for 30 min and then at 4 °C for 90 min. This Sae2 species is denoted as "Sae2 λ". To phosphorylate Sae2 in vitro, Sae2 λ was incubated with 15 U of recombinant human CDK1/cyclin B (New England Biolabs) for 30 min at 30 °C and added to the nuclease reaction mixture. The various Sae2 mutants were obtained by mutagenesis of the *SAE2* gene in the pFB-MBP-Sae2-his plasmid using the QuikChange II XL Site-directed Mutagenesis Kit (Agilent) according to the manufacturer's recommendations. Multiple mutant combinations were obtained upon gene synthesis (GenScript). All Sae2 variants were then expressed in *Sf*9 cells and purified as described above, with the whole procedure scaled down proportionally. The yield of pSae2 mutants, prepared from 250 ml cell culture volume, was ~ 400–800 µg.

Recombinant wt Rad50-Flag, K40A, and K81I (Rad50S) variants were expressed in *Sf*9 cells using pFB-Rad50-FLAG vectors and derivatives. Soluble extracts from 800 ml cells were obtained as for Sae2[24] but with a lysis buffer lacking ethylenediaminetetraacetic acid (EDTA) and containing only 0.5 mM β-mercaptoethanol to avoid reduction of the anti-FLAG antibody. Soluble extracts were incubated with 0.6 ml of anti-FLAG M2 affinity gel (Sigma, A2220) for 1 h at 4 °C. First, the resin was extensively washed with ~150 ml of "High salt" wash buffer containing 50 mM Tris-HCl pH 7.5, 0.5 mM β-mercaptoethanol, 300 mM NaCl, 0.5 mM phenylmethylsulfonyl fluoride, 10% (v/v) glycerol, and 0.1% (v/v) NP40. The resin was then additionally washed with ~50 ml of "Low salt" wash buffer (same as "High salt" buffer, but without NP40 and with only 150 mM NaCl). The proteins were eluted with "Low salt" wash buffer supplemented with 200 µg/ml of 3X FLAG peptide (Sigma).

**DNA substrates**. The 50 and 70 bp DNA substrates were labeled at the 3' end with terminal deoxynucleotidyl transferase (New England Biolabs) and [α-$^{32}$P]-cordycepin 5'-triphosphate (Perkin Elmer)[24]. The oligonucleotides used for the 100 bp dsDNA were labeled identically and included Bio100 (G**T**AAGTGCCGC GGTGCGGGTGCCAGGGCGTGCCCTTGGGCTCCCCGGGCGCGTACTCCAC CTCA TAATCTTCTGCCATGGTCGTAGCAGCCTCCTGCATC) and Bio100c (GATGCAGGAGGCTGCTACGACCATGGCAGAAGATTATGAGGTGGAGTA CGCGCCCGGGGAGCCCAAGGGCACGCCCTGGCACCCGCACCGCGGCAC TTAC). **T** in bold indicates the position of the internal biotin label.

**Electrophoretic mobility shift assay**. The electrophoretic mobility shift assay to determine the DNA-binding capacity of Sae2/pSae2 and the MRX complex was carried out with a 15 µl total reaction volume in a binding buffer containing 25 mM Tris-acetate, pH 7.5, 1 mM dithiothreitol, 2 mM magnesium acetate, 0.1 mg/ml bovine serum albumin (New England Biolabs), and 1 nM (in molecules) DNA substrates. In reactions where MRX was present, 1 mM ATP was added to the buffer. Where indicated, streptavidin (Sigma, 30 nM) was added to the mixture and preincubated for 5 min at room temperature. Purified proteins were then added on

ice and incubated for the indicated times at 30 °C. The complexes were mixed with 5 µl loading dye (50% glycerol, 0.1% bromophenol blue), loaded immediately on native polyacrylamide gel (6% or 4%, as indicated, ratio of acrylamide:bisacrylamide 19:1), and separated by electrophoresis in TBE buffer (89 mM Tris base, 89 mM boric acid, 2 mM EDTA) on ice. Gels were dried on DE81 chromatography paper (Whatman), exposed to a phosphor screen, and scanned using a Typhoon Imager (GE Healthcare). Data were quantitated based on the disappearance of the substrate band using the ImageQuant software. Error bars show standard error of the mean. Two-tailed $t$ test was used to calculate $p$ values, where indicated. Differences between two data populations with $p$ values of >0.05 were not considered significant.

**Size exclusion chromatography**. The gel filtration chromatography was carried out using a Superose 6 10/300 GL column (GE Healthcare) on an ÄKTA prime system (GE Healthcare). The separation buffer contained 50 mM Tris-HCl, pH 7.5, 150 mM NaCl, 10% glycerol, and 0.5 mM β-mercaptoethanol. The recombinant Sae2/pSae2 variant (~75 µg) was diluted into 200 µl of separation buffer, supplemented with 1 mM manganese chloride, and either treated with λ phosphatase (4'800 U, New England Biolabs) or mock-treated (without λ phosphatase) for 90 min at 30 °C. The sample (200 µl) was then loaded onto the gel filtration column and separated at 0.2 ml/min at 4 °C. Owing to the low number of aromatic amino acids in Sae2, the protein was monitored by absorbance at 215 nm.

**Nuclease assays**. Nuclease assays (15 µl volume) were carried out with oligonucleotide-based substrates (1 nM molecules) in a buffer containing 25 mM Tris-acetate, pH 7.5, 1 mM dithiothreitol, 5 mM magnesium acetate, 1 mM manganese acetate, 1 mM ATP, 80 U/ml pyruvate kinase (Sigma), 1 mM phosphoenolpyruvate, and 0.25 mg/ml bovine serum albumin (New England Biolabs). Streptavidin (Sigma), 15 nM for the 50-bp substrate and 30 nM for the 70 bp substrate, was added to the substrate containing reaction mixture and incubated for 5 min at room temperature. Recombinant proteins were then added on ice and incubated for 30 min at 30 °C. Reactions were terminated with stop buffer (0.5 µl 10% [w/v] sodium dodecyl sulfate and 0.5 µl 0.5 M EDTA) and 0.5 µl proteinase K (14–22 mg/ml, Roche), and incubated for 30 min at 37 °C. Terminated reaction mixtures were mixed with an equal volume of loading dye (95% formamide, 20 mM EDTA, 1 mg/ml bromophenol blue) and separated by electrophoresis in 15% denaturing polyacrylamide gels (acrylamide:bisacrylamide, 19:1, Biorad) in TBE buffer. The gels were fixed for 30 min at room temperature in 40% methanol, 10% acetic acid, and 5% glycerol; dried on 3MM paper (Whatman); and exposed to storage phosphor screen (GE Healthcare). The screens were scanned by a Typhoon Imager (GE Healthcare). Images were quantitated using the ImageQuant software. First, DNA cleavage in MRX-only lanes (without Sae2) was removed as a background from all lanes to score specifically for effects of Sae2. Subsequently, the Sae2-dependent DNA cleavage was calculated as products/(substrate + products) in each lane. Error bars show standard error of the mean. Two-tailed $t$ test was used to calculate $p$ values, where indicated. Differences between two data populations with $p$ values of >0.05 were not considered significant.

**Protein interaction assays**. To analyze the binding of Sae2/pSae2 to MRX, MBP-tagged Sae2 was expressed in *Sf*9 cells. Two separate cultures were made. The first cell pellet was obtained from cells upon treatment with phosphatase inhibitors as described above, while for the second pellet, phosphatase inhibitors were omitted. The respective cell extracts (with or without phosphatase inhibitors) were bound to 50 µl amylose resin (New England Biolabs) for 1 h at 4 °C as described in the preparation of recombinant proteins section. The resin was then washed batchwise 5 times with 1 ml wash buffer (50 mM Tris-HCl, pH 7.5, 2 mM EDTA, 80 mM NaCl, 0.2% NP40). After the last wash, the resin with the sample prepared without phosphatase inhibitors was incubated with 800 U of λ phosphatase (New England Biolabs) in the recommended buffer for 15 min at 30 °C, centrifuged briefly, and the supernatant was removed, while the second sample was mock-treated (without λ phosphatase). Both resins were then incubated with 5 µg recombinant MRX for 1 h at 4 °C and washed 4 times with 1 ml wash buffer as described above. Proteins were eluted with 100 µl wash buffer containing 20 mM maltose. To cleave the MBP tag, the eluates were incubated for 1 h at 4 °C with 2 µg of Prescission protease. Protein complexes were separated by sodium dodecyl sulfate-polyacrylamide gel electrophoresis (SDS-PAGE) and detected with silver staining. The same procedure was used to test the binding of Sae2 to 1 µg of recombinant Mre11. Avidin was added to the eluates as a protein stabilizer (100 ng/µl, Sigma), and western blot with anti-His tag antibody (MBL, D291–3, dilution 1:1000) was used for detection.

To analyze the binding of the C-terminal fragment of Sae2 to Rad50 and its variants, the MBP-pSae2 ΔN169 was first partially purified on amylose resin, eluted, and cleaved with PreScission protease. The phosphorylated fragment was then bound to NiNTA resin in 50 mM Tris-HCl, pH 7.5, 2 mM EDTA, 80 mM NaCl, 0.2% NP40, and 10 mM imidazole and washed 5 times with 1 ml of the same buffer. Half of the resin was then incubated with λ phosphatase for 30 min at 30 °C, while the other half was mock-treated. (The amount of λ phosphatase was determined in a separate experiment by monitoring pSae2 ΔN169 dephosphorylation by its mobility in polyacrylamide gel). After a short spin and removal of the supernatant, the resins were incubated with 1 µg of purified Rad50

(or its variants) for 1 h at 4 °C. Resins were washed 6 times with 1 ml buffer as above and the protein complexes were eluted with the same buffer containing 300 mM imidazole. The eluates were supplemented with Avidin to stabilize the proteins (Sigma, 100 ng/µl) and analyzed by Ponceau staining or western blotting (anti-Rad50 antibody, Thermo Scientific, PA5–32176, dilution 1:1000). To perform the experiment reciprocally, Rad50-FLAG was expressed in *Sf*9 cells and the cell lysate was bound to anti-FLAG M2 affinity gel (Sigma, A2220) for 1 h at 4 °C as described in the preparation of recombinant protein section. The resin was washed 5 times with 1 ml wash buffer (50 mM Tris-HCl, pH 7.5, 2 mM EDTA, 150 mM NaCl, 0.2% NP40) and incubated for 1 h at 4 °C with 1 µg of either mock-treated or dephosphorylated pSae2 ΔN169. The resin was then washed 6 times with 1 ml wash buffer and the protein complexes were eluted with wash buffer supplemented with 200 µg/ml 3X FLAG peptide (Sigma). Avidin was added to the final eluate (100 ng/µl) as a stabilizer, and the protein complexes were analyzed by Ponceau staining and western blotting with anti-His tag antibody (MBL, D291–3, dilution 1:1000).

**Transmission electron microscopy**. Non-sized Sae2 and pSae2 samples were diluted to 800 nM in EM buffer (20 mM HEPES, 130 mM NaCl, 2% glycerol, and 0.5 mM DTT, pH 7.55) and 4 µl were applied to carbon film 300 mesh copper grids (CF300-CU from Electron Microscopy Sciences, 215–412–8400) that were glow discharged for 45 s with 25 mA using the Emitech K100X glow discharge system under conditions that create a negatively charged grid surface. After incubation of 1 min on the grid, the excess sample was manually blotted with Whatman filter paper, washed twice with EM buffer, and stained with two drops of 2% uranyl acetate. Transmission electron micrographs were acquired with a FEI Morgagni 268 microscope at 100 kV using a CCD 1376 × 1032 pixel camera at a nominal defocus of −6 µm (Fig. 1i, j) or with a FEI Tecnai F20 FEG microscope at 200 kV using a Falcon II 4 K Direct Electron Detector operated in low-dose mode and a nominal defocus of −2 µm (Supplementary Fig. 2b, c).

**Atomic force microscopy**. In all, 50 nM hyperphosphorylated pSae2 in 40 mM Hepes pH 7, 5 mM MgCl₂, 25 mM NaCl, 0.1 mM Tris(2-carboxyethyl)phosphine was deposited on freshly cleaved mica (Ted Pella, Inc.) for <1 min and then rinsed with 800 µl of molecular biology grade water. The mica surface was dried by blotting and a nitrogen stream. Images were captured in air with a Nanoscope IIIa (Digital Instruments) microscope in tapping mode using PointProbe®Plus-NCL silicon probes (Nanosensors). Images were collected at a size of $1 \times 1$ µm². Image Metrics v 1.13 (in-house Erie lab) was used for image processing and particle volume analysis. Standard image processing utilized plane subtraction and flattening. Statistical analyses were carried out using GraphPad Prism 7.0 (GraphPad Software Inc.). The molecular weight of pSae2 was derived from a Gaussian distribution of volumes as determined by ImageMetrics and a published standard curve[31].

**Mass spectrometry**. Purified pSae2 was run on a SDS-PAGE gel, stained with Coomassie blue, and excised from the gel. Gel pieces were reduced with 10 mM Tris(2-carboxyethyl)phosphine, alkylated with 20 mM iodoacetamide, and cleaved with 0.1 µg porcine sequencing grade trypsin (Promega) in 25 mM ammonium bicarbonate, pH 8.0, at 37 °C for 16 h. The extracted peptides were analyzed by capillary liquid chromatography tandem mass spectrometry with an EASY-nLC 1000 using the two-column set-up (Thermo Scientific). The peptides were loaded in 0.1% formic acid, 2% acetonitrile in water onto a peptide trap (Acclaim PepMap 100, 75 µm×2 cm, C18, 3 µm, 100 Å) at a constant pressure of 600 bar. Peptides were separated at a flow rate of 200 nl/min with a linear gradient of 2–30% buffer B in buffer A in 20 min, followed by a linear increase from 30% to 50% in 5 min, 50–80% in 2 min, and the column was finally washed for 14 min at 80% B (Buffer A: 0.1% formic acid, buffer B: 0.1% formic acid in acetonitrile) on a 75 µm×15 cm ES800 C18, 2 µm, 100 Å column mounted on a DPV ion source (New Objective) connected to a Orbitrap Velos (Thermo Scientific). The data were acquired using 60,000 resolution for the peptide measurements in the Orbitrap and a top 5 method with a CID fragmentation for each precursor and fragment measurement in the LTQ. Mascot Distiller 2.5 and MASCOT 2.5 searching the yeast sub set of the swissprot version 2015_01 DB was used to identify the phospho peptides. The enzyme specificity was set to trypsin allowing for up to three incomplete cleavage sites. Carbamidomethylation of cysteine (+57.0245) was set as a fixed modification, phosphorylation of serine, threonine and tyrosine (+79.9663 Da) oxidation of methionine (+15.9949 Da), and acetylation of protein N-termini (+42.0106 Da) were set as variable modifications. Parent ion mass tolerance was set to 5 ppm and fragment ion mass tolerance to 0.6 Da. The results were validated with the program Scaffold Version 4.4 and ScaffoldPTM (Proteome Software). Peptide identifications were accepted if they could be established at >50.0% probability as specified by the Peptide Prophet algorithm[45,46], and phosphorylation sites were accepted if they had a >80% site probability as calculated with ScaffoldPTM.

**Spo11-oligonucleotide formation assay**. In vivo nuclease activity of the MRX-Sae2 complex was assessed via the release of Spo11 from meiotic DSB ends[14,34,38]. Briefly, *S. cerevisiae* cells were pregrown in YPA medium for 14–16 h (containing 300 µg/ml hygromycin to maintain CEN-plasmids), then washed, and resuspended

in 1% potassium acetate with nutritional supplements. At the indicated time points, culture samples (10–20 ml) were harvested, lysed in the presence of 10% tri-chloroacetic acid, and resulting precipitates were dissolved in 2% SDS, 0.5 M Tris-HCl, pH 8.1, and 1 mM EDTA. Soluble material was diluted four-fold to a final concentration of 1% Triton, 0.5% SDS, 140 mM Tris pH 8.1, 150 mM NaCl, 1 mM EDTA, and incubated with mixing for >4 h at 4 °C with anti-FLAG antibody (#F1804, Sigma-Aldrich) and Protein-G-agarose (#05015979001 Roche/Sigma-Aldrich). Immune complexes were collected by centrifugation, washed, equilibrated in 1× New England Biolabs Terminal transferase buffer, and incubated with 10 U terminal deoxynucleotidyl transferase (#EP0161 Fermentas/ThermoFisher) and 5 µCi alpha-$^{32}$P dCTP (Perkin Elmer) for 1 h at 37 °C. Labeled Spo11-oligo complexes were washed, denatured in loading buffer, resolved by 8% SDS-PAGE, and detected via phosphorimaging using a Fuji FLA5100 scanner and ImageGauge software. Where indicated, quantitation is shown below the lines as an average (relative to wt) ± coefficient of variation (standard deviation/mean) and the number of independent measurements (n).

**Yeast strains and plasmids.** S. cerevisiae strains used are derivatives of the SK1 genetic background. sae2Δ::KanMX and tel1Δ::HphNT2 are partial gene deletions generated via LiAc transformation and are functional knockouts[38,47]. The clb2-HA-MEC1::KanMX construct places MEC1 under the control of the mitosis-specific CLB2 promoter and was generated via PCR-mediated transformation[48]. The centromeric vectors were generated by cloning the SAE2 open reading frame including 500 bp upstream and downstream into a derivative of pRS416 in which the HphNT2 marker was cloned at the MCS. The HA3 in-frame tag was cloned into an NheI site generated at the C-terminus of SAE2 by site-directed mutagenesis. Individual point mutations in the SAE2 ORF were introduced via site-directed mutagenesis using standard methods. Inducible SAE2 constructs were generated by cloning 502 bp of the SAE2 5' untranslated region (UTR) and 1192 bp (or truncated fragments) of the SAE2 ORF plus 3' UTR into plasmid pYM_N23[49] flanking the GAL1 promoter. Point mutations were introduced by site-directed mutagenesis. Fragments were liberated by restriction digestion and transformed into host strains harboring the Gal4-ER fusion construct inserted at the URA3 locus[48], thereby modifying the natural SAE2 locus. The base SK1 haploid genotype is: ho::LYS2, lys2, arg4-nsp, leu2::hisG, his4X::LEU2, nuc1::LEU2, SPO11-FLAG3His3::KanMX. A complete list of all strains, plasmids, mutagenesis primers, and construction details is available on request.

## Data availability

All relevant data are available from the corresponding authors upon reasonable request.

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

## Acknowledgements

We thank members of the Cejka and Neale laboratories for critical comments on the manuscript and Alessandro Sartori (University of Zurich) for the Superose 6 column and the ETH ScopeM facility for help with the EM analysis. This work was supported by the Swiss National Science Foundation (SNCF) Grants 31003A_175444 and PP00P3_159323 to P.C., European Research Council (ERC) Grant 681630 to P.C., ERC Grant 311336 to M.J.N., BBSRC grant BB/M010279/1 to M.J.N., Royal Society grant UF110009 to M.J.N., Wellcome Trust grant 200843/Z/16/Z to M.J.N., US National Institute of Health Intramural Program-US National Institute of Environmental Health Sciences (NIEHS) grant 1Z01ES102765 to R.S.W. J.K.R. was supported by a Boehringer Ingelheim Fonds PhD fellowship. The laboratory of N.H.T. is supported by the Novartis Research Foundation. V.K. was supported by the Helmut Horten Foundation, and work in the M.P. laboratory was funded by the SNSF, ERC, and the ETH Zürich.

## Author contributions

E.C. and P.C. designed and performed the biochemical experiments. D.J., V.G., and M.J.N. designed the genetic experiments, and D.J. performed the genetic experiments. S.N.A., D.A.E., and R.S.W. designed, performed, and analyzed the atomic force microscopic experiments. V.K., R.I.E., and M.P. designed and performed transmission electron microscopic analysis. J.K.R., D.H., and N.H.T. designed and performed mass spectrometric experiments. E.C., P.C., and M.J.N. wrote the manuscript. All authors provided comments.

## Additional information

**Competing interests:** The authors declare no competing interests.

