## [Peer Review File · Nature Communications]

Reviewers' comments:

Reviewer #1 (Remarks to the Author):

Here, Cannavo, Johnson, Cejka et al investigate of the ability of Sae2 to activate the endonuclease activity of the MRX complex at DNA double-strand breaks with blocked ends. These proteins constitute the catalytic core of the ensemble that processes DNA ends in preparation of repair by homologous recombination in mitotic and meiotic cells. The Cejka group has already published a series of elegant studies defining the end resection roles of MRX-Sae2 and its human equivalent MRN-CtIP, and this manuscript significantly advances their previous work. In short, the authors show that Sae2 phosphorylation affects resection by two distinct mechanisms, viz., that it regulates Sae2 oligomerization and also regulates the interaction between Sae2 and MRX. The relevance of phosphorylation sites is dissected, leading to several novel findings. A previously unknown interaction between the phosphorylated Sae2 C-terminus and Rad50 is identified, and the rad50S mutant is revealed to be deficient in this interaction, thus providing a gratifying explanation to a longstanding enigma in the field. In all, the authors identify 22 functionally relevant phosphorylation sites in Sae2!

This is an impressive paper that resolves questions about MRX-Sae2 function and makes an important contribution toward establishing the large body of Cejka's work as the "gold standard" in studies of DNA resection in eukaryotes.

The paper will be eminently suited to Nature Communications after revisions:

1. It is curious that Sae2 phosphorylation reduces DNA binding, yet increases MRX stimulation. Can the authors offer an explanation for this? Is the reduced DNA binding a nonspecific consequence of increasing the negative charge on Sae2? Does it actually prompt the turnover of the ensemble with MRX? A thoughtful discussion in these regards would be most welcome.
2. In Figure 1G, the DNA substrate used should be indicated either in the figure or in the legend. Is Figure 1G the quantitation of the gel shown in Figure S1A?
3. On p. 6 the authors write "Dephosphorylation of pSae2 with λ phosphatase resulted in the formation of soluble multimers that shifted the polypeptide complex into the void volume..." It seems to me that dephosphorylation produces two distinct species, a multimeric form in the void and a monomeric form. As written, the sentence implies that the monomeric form is responsible for both the peak in the void and the peak in fractions 18-19.
4. Is there evidence that S289 is actually phosphorylated in cells? It is odd that phosphorylation at some Mec1/Tel1 target SQ motifs stimulate Sae2's ability to activate MRX whereas this one site inhibits. The observation that mec1 and tel1 strains are deficient at Spo11 DSB processing suggests that these kinases are mainly stimulatory. An alternative explanation for the data could be that the effect of the D/E mutants at S289 are due to misfolding of the protein rather than a phospho-mimic effect.
5. In the paragraph on Page 11 addressing the N123A R127A mutant, the authors should mention whether they tested this variant, the wild-type version, and the phospho version of Sae2 for nuclease activity in vitro. The reported nuclease activity of this protein is controversial (and not believable, in my view) and any data that would clarify this point should be shown.

Reviewer #2 (Remarks to the Author):

The authors have delineated how phosphorylation of Sae2 controls the initiation of DNA end resection. The authors provide evidence that phosphorylation at multiple sites on Sae2 promotes the formation of Sae2 tetramers. Phosphorylation of Sae2 regulates the interaction with Rad50 that lead to activation of the Mre11 nuclease. The authors use a combination of biochemistry and biophysical imaging to show that phosphorylation of Sae2 promotes the formation of oligomers whereas non-phosphorylated Sae2 aggregates or forms multimers. The authors used both truncation variants of Sae2 and a rather impressive array of point mutants to decipher the role of phosphorylation in the function of Sae2. Impressively, the authors made a variant of Sae2 that contained 22 amino acid substitutions in an attempt to greatly reduce phosphorylation of Sae2. Lastly, the authors help to shed light on a 28 year old mystery; the Rad50S K81L separation of function mutant of RAD50. In agreement with a previous report by Cejka and separately the Sung lab, the authors add more evidence to refute the erroneous claim that Sae2 itself is a nuclease.

This manuscript is well written. The data, for the most part, is of outstanding quality. There are a number of minor concerns that the authors should address and clarify in the manuscript. My already high enthusiasm for this manuscript would be strengthened by addressing the concerns below. The findings in the manuscript are highly novel and very thorough. The results extend the knowledge of DNA end resection significantly. The findings of this paper will be read and cited by many researchers in the DNA repair and meiosis field, including myself.

Concerns

1. Throughout the manuscript, P-values would greatly strengthen the data. An N = 2 is not a typically accepted number of replicates to instill confidence in the data.
2. The authors did an exceptional job to purify what appears to be >95% phosphorylated or >95% unphosphorylated Sae2. The authors are to be commended for including weakly phosphorylated Sae2 purified from a previous procedure to validate both previous and current results. That said, Coomassie stain is not the most sensitive. Do the authors see the same level of purity using silver stain? Why do the authors require 280 U of lambda phosphatase to dephosphorylate 1.3 micrograms of pSae2? Perhaps this amount of phosphatase a typo?
3. The first sentence describing DSB formation should include stalled or damaged replication forks.
4. Figure 1. The panels show a gel in panel E that has no detectable amounts of cleavage. The authors on page 5 in results state that 'treatment of pSae3 with lambda phosphatase almost completely eliminated its....' However, the graph in panel F has approximately 8-10% cleavage (which is not 'almost completely eliminated') with experiments in excess of N=3. The authors should reconcile these to be consistent. The data in panel F would be greatly strengthened with p-values.
5. In Figure 1H, the dephosphorylated pSae2 is forming multimers and potentially aggregates (multimers). How do the authors know that the enhanced DNA binding of dephosphorylated Sae2 is not due to aggregation of the dephosphorylated Sae2? Furthermore, the binding site size of Sae2 may change as a result of the phosphorylation status of Sae2. Do the authors know the binding site size of the different variants of Sae2?
6. Supp fig 1 The data in C and E would be greatly strengthened with p-values. Especially in panel E, considering the difference between MRX and MRX plus either pSae2 and pSae2 lambda is not that

great in panel D compared to panel B. The quantitation of panel D in panel E seems to be a little off. This is likely due to the streaking of the substrate in the gel that is not present in panel B. I assume this is due to the streptavidin?

7. On page 5, the authors state the 'hyperphosphorylated pSae2 had a lower DNA binding affinity compared to the Lambda phosphatase treated variant'. The data as presented is suggestive of this conclusion. As noted above, there could be a different reason for the apparent affinity changes in Sae2, ie. aggregation or binding site changes due to phosphorylation. The authors should address the possibility that aggregation is causing the improved DNA binding activity for the dephosphorylated Sae2.

8. The authors indicate an N =2 for the EMSA experiments. An N = 2 is generally not acceptable for any meaningful statistics. In most of the figures, the authors indicate an N > or = 2. I am sure the authors know specifically how many replicates they did for each experiment. They should specifically state this information.

9. I assume the authors performed the size exclusion experiments with Sae2 variants devoid of a fused MBP. In the methods, the authors note that Prescission protease is used after elution from the amylose resin. Was the MBP removed from this cleaved dephosphorylated Sae2? Where did the cleaved MBP go since there is not a peak around 45-50 kDa for MBP in the size exclusion experiments and only one for lambda phosphatase. The authors should clarify the methods and consider indicating in the text when MBP-Sae2 is used and when a cleaved Sae2 is used.

10. Is it possible that the large aggregates/oligomers of Sae2 that form are due to the higher concentrations of Sae2? Also, there is DNA present in Figure 1F and not in the size exclusion experiments. It is very likely that Sae2 whether phosphorylated or not will take on conformational changes when bound to DNA. The authors need to reconsider their conclusion on page 15 that pSae2 tetramers represent the active pSae2 species that optimally promote the Mre11 nuclease within the MRX complex. The authors did not perform the size exclusion chromatography with DNA present whereas the nuclease experiment obviously did. Therefore, the comparison between the DNA cleavage assay and the oligomeric state of Sae2 does not seem to be appropriate without further clarification.

11. The use of AFM to estimate the formation of a tetramer is very nice.

12. Although qualitative, the authors use of TEM is also nice to see. The authors state that non-sized Sae2 was used in the TEM experiments which yielded mix populations of species of Sae2 to be visualized. These results are in agreement with the size exclusion experiments. That said, it is not clear why the authors did not evaluate the sized Sae2 variants. This would have clearly confirmed the which species of Sae2 is forming aggregates/multimers and oligomers.

13. Did the authors visualize the different species of Sae2 on DNA?

14. It is possible that the phosphorylation status of Sae2 may have altered the way Sae2 interacts with the carbon film in the TEM. For instance it may have flattened out. To this end, the authors mentioned that the AFM images were corrected for flattening. This suggests that the different phosphorylation states of Sae2 may have altered the way Sae2 interacts with the freshly cleaved mica. The authors should comment on these issues more than is currently in the manuscript.

15. In Figure 2 panel D, why is there not a lambda phosphatase band present? If the molecular weight markers are correct, it should be present on the gel as it is in every gel in the manuscript. Is the lambda phosphatase masked by the Sae2-delta169 variants?

16. Figure 3, the quantitation in B is not in agreement with the respective panels in A.
17. Figure 3, the quantitation in G is not in agreement with the respective panel in F.
18. Figure 4, the quantitation in L is not in agreement with the respective panels in K.
19. On page 9, the authors state that pSae2 that had been dephosphorylated on page 9, the authors state that pSae2 that had been dephosphorylated by lambda phosphatase can be partially activated upon phosphorylation by human CDK1/Cyclin B. Perhaps the authors included the wrong panel for F in figure 3, because the gel has no appreciable change in the cleavage pattern. This is in stark contrast to the graph in panel G. Which is correct?
20. Since the authors point out on page 15 that ATP hydrolysis by Rad50 is necessary for the MRX-Sae2 endonuclease, does unphosphorylated or phosphorylated Sae2 influence the ATPase activity of Rad50?

Reviewer #3 (Remarks to the Author):

Sae2 is a DNA repair enzyme conserved across most species. Its major role is regulation of Mre11-Rad50-Xrs2 complex, a key multifunctional enzyme in DNA damage response. One of the major functions of MRX complex is in resection of 5' ends of DNA double strand breaks to form 3' ssDNA, essential process for recombination and DNA damage checkpoint. The authors of this manuscript focus on the control of MRX activity in resection by Sae2. Specifically they look at the role of Sae2 phosphorylation. They purified Sae2 from Sf9 cells in the presence of phosphatase inhibitors. Purified this way Sae2 was phosphorylated to much higher level than observed before and showed increased activities meaning higher stimulation of Mre11 nuclease activity. They provide convincing evidence that phosphorylation of Sae2 regulates oligomerization of Sae2. Hyperphosphorylated form of Sae2 forms specific oligomeric form (tetrameric) while nonphosphorylated form is observed mostly as multimeric. The domains and phosphorylation sites needed for the formation of tetrameric form and the importance of tetrameric form are defined. The authors also carefully examine the function of Mec1 and Tel1 mediated phosphorylation and Cdk1 mediated phosphorylation of Sae2 in stimulating MRX activity and in formation of most active tetrameric form. They performed mass spec to identify additional phosphorylation sites and document that Sae2 phosphorylation at C-terminus is important for optimal activity of Sae2.

This is the most comprehensive study of Sae2 phosphorylation. It clarifies some important controversies (oligomerization, nuclease activity) and provides new mechanistic insights on how phosphorylation of Sae2 stimulates MRX activity. It also opens a lot of questions for the future studies on Sae2. Considering that MRX and Sae2 play central role in damage response and recombination this work deserves strong consideration. The manuscript is well written.

Minor questions/concerns that authors could discuss.

Is there a possibility that hyperphosphorylated form of Sae2 as purified from Sf9 cells is not the form that exist normally in yeast. Perhaps phosphatases eliminate most of the phosphate groups in cells or kinases in Sf9 cells have different specificity. It seems that besides Cdk1 and Tel1/Mec1, other kinases are involved as many sites tested here and shown to be important for optimal activity are not within the consensus motifs of these kinases. The authors should comment on this. Is there any order of phosphorylation events (one phosphorylation is needed to observe another one?). Are all phosphorylation events controlled in cell cycle? It seems that Cdk1 sites within Sae2 are not needed for other Sae2 phosphorylation events, so how other phosphorylation events are controlled in cell

cycle?

Other points

Page 9: "The ~5-fold decrease in MRX-dependent DNA cleavage upon dephosphorylation of pSae2 S267E demonstrated that other sites in pSae2, in addition to S267, must be phosphorylated for its optimal stimulation of the MRX endonuclease."

It's a bit too strong conclusion. The alternative view is that phosphomimetic form of Sae2 only partially restores the activity of Sae2. Phosphomimetic forms of proteins often have intermediate level of activity in cells. In general the authors should consider this possibility when testing pseudo phosphorylated forms of Sae2.

Abstract

"The lack of this interaction explains the phenotype of rad50S mutants defective in the processing of Spo11-bound DNA ends during meiotic recombination."

It would be great to introduce rad50S better or rephrase the statement for general audience.

In the model on Figure 7, It would be good the add the names of kinases involved.

Page 12

"For subsequent analysis, we selected 22 putative pSae2 phosphorylation sites based on our mass-spectrometry analysis and previous work; 15 of those pSae2 sites had been previously found to be modified in vivo"

Many of these published sites are coming from high throughput studies and were not confirmed. It's a bit overstatement to say these all sites are phosphorylated in cells. It would good to add information how these sites were identified.

We would like to thank the reviewers for their time and interest reading our manuscript, as well as for providing helpful and constructive suggestions. Please find below how we addressed their comments.

Reviewer #1 (Remarks to the Author):

Here, Cannavo, Johnson, Cejka et al investigate of the ability of Sae2 to activate the endonuclease activity of the MRX complex at DNA double-strand breaks with blocked ends. These proteins constitute the catalytic core of the ensemble that processes DNA ends in preparation of repair by homologous recombination in mitotic and meiotic cells. The Cejka group has already published a series of elegant studies defining the end resection roles of MRX-Sae2 and its human equivalent MRN-CtIP, and this manuscript significantly advances their previous work. In short, the authors show that Sae2 phosphorylation affects resection by two distinct mechanisms, viz., that it regulates Sae2 oligomerization and also regulates the interaction between Sae2 and MRX. The relevance of phosphorylation sites is dissected, leading to several novel findings. A previously unknown interaction between the phosphorylated Sae2 C-terminus and Rad50 is identified, and the rad50S mutant is revealed to be deficient in this interaction, thus providing a gratifying explanation to a longstanding enigma in the field. In all, the authors identify 22 functionally relevant phosphorylation sites in Sae2!

This is an impressive paper that resolves questions about MRX-Sae2 function and makes an important contribution toward establishing the large body of Cejka's work as the "gold standard" in studies of DNA resection in eukaryotes.

The paper will be eminently suited to Nature Communications after revisions:

1. It is curious that Sae2 phosphorylation reduces DNA binding, yet increases MRX stimulation. Can the authors offer an explanation for this? Is the reduced DNA binding a nonspecific consequence of increasing the negative charge on Sae2? Does it actually prompt the turnover of the ensemble with MRX? A thoughtful discussion in these regards would be most welcome.

Answer: It is possible that the reduction of DNA binding capacity is simply a charge issue. We hypothesize that DNA binding by Sae2 is not required for the stimulatory effect on the MRX endonuclease. In addition to the paradox mentioned by the reviewer, we note that DNA cleavage positions are determined already by the MRX complex, and Sae2 only stimulates the efficiency. We mention this more clearly in the revised text (Discussion). Our thought is that DNA binding by Sae2 may reflect another of its functions that is separate from being a co-factor of the Mre11 nuclease. Please see also the answer to reviewer #2 below.

2. In Figure 1G, the DNA substrate used should be indicated either in the figure or in the legend. Is Figure 1G the quantitation of the gel shown in Figure S1A?

Answer: Thank you, we indicate the substrate (100 bp-long dsDNA) in the Figure legend and Y axis. The quantitation is based on experiments such as in S1A (the image shown is one of those, i.e. a representative experiment). This is also mentioned in the legend.

3. On p. 6 the authors write "Dephosphorylation of pSae2 with λ phosphatase resulted in the formation of soluble multimers that shifted the polypeptide complex into the void volume..." It seems to me that dephosphorylation produces two distinct species, a multimeric form in the void and a monomeric form. As written, the sentence implies that the monomeric form is responsible for both the peak in the void and the peak in fractions 18-19.

Answer: The 18-19 fraction is the lambda phosphatase, not Sae2. We now mention this both in the text (when we describe Fig. 1h) and all respective Figure legends to avoid confusion.

4. Is there evidence that S289 is actually phosphorylated in cells? It is odd that phosphorylation at some Mec1/Tel1 target SQ motifs stimulate Sae2's ability to activate MRX whereas this one site inhibits. The observation that mec1 and tel1 strains are deficient at Spo11 DSB processing suggests that these kinases are mainly stimulatory. An alternative explanation for the data could be that the effect of the D/E mutants at S289 are due to misfolding of the protein rather than a phospho-mimic effect.

Answer: The reviewer is correct, and we now mention this possibility in the text. We do not have evidence that this site is modified in cells, and therefore we cannot exclude that the effects of D/E substitutions compared to A is due to protein misfolding. Alternatively, the E/D substitutions may cause protein misfolding, and it therefore remains to be established whether S289 phosphorylation occurs as a regulatory mechanism in cells.

5. In the paragraph on Page 11 addressing the N123A R127A mutant, the authors should mention whether they tested this variant, the wild-type version, and the phospho version of Sae2 for nuclease activity in vitro. The reported nuclease activity of this protein is controversial (and not believable, in my view) and any data that would clarify this point should be shown.

Answer: We never prepared the recombinant N123A R127A Sae2 mutant. We show below that the hyperphosphorylated, weakly phosphorylated and λ phosphatase-treated Sae2 variants (240 nM, i.e. 240-fold higher than the DNA concentration, Figure R1) displayed no detectable nuclease activity when assayed on a 3' labeled Y-structure DNA, while 15-fold lower phosphorylated Sae2 concentrations resulted in >50% dsDNA cleavage in conjunction with MRX (e.g. Fig. 1 of our manuscript). We include the following note in the manuscript: "We therefore conclude that Sae2 functions in resection as an activator of Mre11, rather than having an intrinsic catalytic function (Cannavo and Cejka, 2014)."

Figure R1. Nuclease assay with Sae2 variants and Dna2. Phosphorylated Sae2, prepared with phosphatase inhibitors (lane 2), weakly phosphorylated Sae2, prepared without phosphatase inhibitors (lane3), Sae2 treated with lambda phosphatase during protein purification (lane 4) and phosphorylated Sae2 treated with lambda phosphatase just prior to experiment (lane 5), as well as Dna2 were used in a nuclease assay. 3'-labeled Y-structure DNA was used a substrate.

[Editorial Note: Unpublished data statement redacted from the Peer Review File by the Editorial Team as per Author request.]

Reviewer #2 (Remarks to the Author):

The authors have delineated how phosphorylation of Sae2 controls the initiation of DNA end resection. The authors provide evidence that phosphorylation at multiple sites on Sae2 promotes the formation of Sae2 tetramers. Phosphorylation of Sae2 regulates the interaction with Rad50 that lead to activation of the Mre11 nuclease. The authors use a combination of biochemistry and biophysical imaging to show that phosphorylation of Sae2 promotes the formation of oligomers whereas non-phosphorylated Sae2 aggregates or forms multimers. The authors used both truncation variants of Sae2 and a rather impressive array of point mutants to decipher the role of phosphorylation in the function of Sae2. Impressively, the authors made a variant of Sae2 that contained 22 amino acid substitutions in an attempt to greatly reduce phosphorylation of Sae2. Lastly, the authors help to shed light on a 28 year old mystery; the Rad50S K81L separation of function mutant of RAD50. In agreement with a previous report by Cejka and separately the Sung lab, the authors add more evidence to refute the erroneous claim that Sae2 itself is a nuclease.

This manuscript is well written. The data, for the most part, is of outstanding quality. There are a number of minor concerns that the authors should address and clarify in the manuscript. My already high enthusiasm for this manuscript would be strengthened by addressing the concerns below. The findings in the manuscript are highly novel and very thorough. The results extend the knowledge of DNA end resection significantly. The findings of this paper will be read and cited by many researchers in the DNA repair and meiosis field, including myself.

Concerns

1. Throughout the manuscript, P-values would greatly strengthen the data. An N = 2 is not a typically accepted number of replicates to instill confidence in the data.

Answer: During revision, we repeated a number of experiments to have at least 3 replicates for each quantitation (in many cases, there are 4-5). Also, we perform each experiment together with relevant standards (i.e., fully phosphorylated wt, dephosphorylated wt Sae2), and analyze multiple protein concentrations, which makes us confident about our data. As we present most results as protein titrations, we found it impractical to calculate p-values for all concentrations, and opted to include those only in selected cases, where we thought they would be most informative. We hope the reviewer will agree.

2. The authors did an exceptional job to purify what appears to be >95% phosphorylated or >95% unphosphorylated Sae2. The authors are to be commended for including weakly phosphorylated Sae2 purified from a previous procedure to validate both previous and current results. That said, Coomassie stain is not the most sensitive. Do the authors see the same level of purity using silver stain? Why do the authors require 280 U of lambda phosphatase to dephosphorylate 1.3 micrograms of pSae2? Perhaps this amount of phosphatase a typo?

Answer: We show below silver stain image of our Sae2 preparations (Figure R2). As can be seen below, when loading 500 ng of Sae2, we start to see minor contaminant bands. Additionally, it can be seen that in our phosphorylated Sae2 variant, there is still a minor fraction of the polypeptide that is not phosphorylated. This is in agreement with our TEM analysis, which detected a fraction of multimers even in the hyperphosphorylated Sae2 population.

Figure R2. Phosphorylated Sae2 (pSae2, 500 ng) and lambda-phosphatase treated Sae2 (Sae2 λ , 500 ng, lambda phosphatase was removed during purification) were separated on a polyacrylamide gel and stained with silver.

According to NEB, "100 units of Lambda PP remove ~100% of phosphates (0.5 nmol) in phosphorylated myelin basic protein (phospho-MyBP, 18.5 kDa) in 30 minutes in a 50 μ l reaction. The concentration of phospho-MyBP is 10 μ M with respect to phosphate."

In our case, considering ~8 phosphorylation sites, we have ~10 μ M phosphate concentration in our dephosphorylation reaction, therefore we are about 3-fold over the recommended amount (corresponding to 0.7 μ l of lambda phosphatase). As we use sub-aliquoted and refrozen lambda phosphatase, we selected this amount to be on a safe side.

3. The first sentence describing DSB formation should include stalled or damaged replication forks.

Answer: Done, thank you.

4. Figure 1. The panels show a gel in panel E that has no detectable amounts of cleavage. The

authors on page 5 in results state that 'treatment of pSae3 with lambda phosphatase almost completely eliminated its....' However, the graph in panel F has approximately 8-10% cleavage (which is not 'almost completely eliminated') with experiments in excess of N=3. The authors should reconcile these to be consistent. The data in panel F would be greatly strengthened with p-values.

Answer: We changed the text into "treatment ... dramatically reduced" to be consistent. We note that an apparent experimental variability comes from the specific activity of our ³²P-labeled substrate. With a freshly labeled substrate, we can detect cleavage activity stimulated by non-phosphorylated Sae2, or in the absence of Sae2 (MRX only), e.g. in Fig. 3c (lane 2). Experiments with less "hot" substrates fail to detect these cleavage products. P-values have been included.

5. In Figure 1H, the dephosphorylated pSae2 is forming multimers and potentially aggregates (multimers). How do the authors know that the enhanced DNA binding of dephosphorylated Sae2 is not due to aggregation of the dephosphorylated Sae2? Furthermore, the binding site size of Sae2 may change as a result of the phosphorylation status of Sae2. Do the authors know the binding site size of the different variants of Sae2?

Answer: The reviewer is correct, please see also our response to Reviewer 1. We do not believe that DNA binding of Sae2 *per se* is essential for the MRX-Sae2 DNA cleavage (see Discussion), and certainly represents an interesting topic for future studies. We note in the revised version of the text that the DNA binding may be affected by charge and/or protein multimerization (see Discussion). "We show that phosphorylation reduces the capacity of pSae2 to bind DNA, which may be due to conformational change or a negative overall charge of the hyperphosphorylated polypeptide. We also cannot exclude that non-phosphorylated or weakly phosphorylated Sae2 aggregates on DNA, which increases its apparent DNA binding activity. However, as DNA cleavage positions are solely determined by the MRX complex and Sae2 only promotes cleavage efficacy, we favor the hypothesis that DNA binding by Sae2, at least in the simple reconstituted system, is not required for the clipping function of MRX."

We note (Figure R3, see below) that the apparent DNA binding affinity is dependent on the substrate length, with longer substrates being bound better than shorter ones. Even very short substrates (20 bp) shift to the wells of our gels when protein-bound, indicating that the site size of Sae2 is either very small, or that that protein forms high order multimers/aggregates on DNA. However, both phosphorylated and non-phosphorylated forms behave identically qualitatively in binding reactions (no distinct intermediate-size bound products). We also note (see below) that a cold competitor added to the binding reaction can displace at > 50% of the dephosphorylated Sae2-DNA complex, showing that the protein bound species are not fully inactivated protein aggregates. As we do not fully understand the DNA binding behavior, and its relevance is not apparent for the clipping reaction, we prefer not to include these results in the manuscript.

Figure R3. DNA binding of phosphorylated (pSae2) and dephosphorylated Sae2 (pSae2 λ). a, Binding to dsDNA substrates of various lengths. b, Dephosphorylated Sae2 was bound to 100 bp-long dsDNA, and subsequently incubated with cold competitor dsDNA (identical 100 bp long dsDNA).

6. *Supp fig 1 The data in C and E would be greatly strengthened with p-values. Especially in panel E, considering the difference between MRX and MRX plus either pSae2 and pSae2 lambda is not that great in panel D compared to panel B. The quantitation of panel D in panel E seems to be a little off. This is likely due to the streaking of the substrate in the gel that is not present in panel B. I assume this is due to the streptavidin?*

Answer: We now include data based on 4 experiments, and include p-values. We describe in methods that the quantification of electrophoretic mobility shift assays was based on the disappearance of the substrate band. We do not find evidence that Sae2 (either form) significantly affects DNA binding of MRX.

Indeed, the streptavidin-bound DNA substrates are more difficult to resolve, and we believe that self-interaction of streptavidin results in the streaking as noted by the reviewer.

7. *On page 5, the authors state the 'hyperphosphorylated pSae2 had a lower DNA binding affinity compared to the Lambda phosphatase treated variant'. The data as presented is suggestive of this conclusion. As noted above, there could be a different reason for the apparent affinity changes in Sae2, ie. aggregation or binding site changes due to phosphorylation. The authors should address the possibility that aggregation is causing the improved DNA binding activity for the*

dephosphorylated Sae2.

Answer: Please see our response above, these possibilities are now discussed in the text.

8. The authors indicate an N =2 for the EMSA experiments. An N = 2 is generally not acceptable for any meaningful statistics. In most of the figures, the authors indicate an N > or = 2. I am sure the authors know specifically how many replicates they did for each experiment. They should specifically state this information.

Answer: We now include quantitation of experiments that have been performed at least 3 times for each data point throughout the manuscript. In many cases there are more repeats. When we have e.g. 4 replicates but one or a few data points in a particular gel cannot be quantitated due to e.g. broken well or debris in a particular lane, we exclude this value. In this case we write N > or = 3.

9. I assume the authors performed the size exclusion experiments with Sae2 variants devoid of a fused MBP. In the methods, the authors note that Prescission protease is used after elution from the amylose resin. Was the MBP removed from this cleaved dephosphorylated Sae2? Where did the cleaved MBP go since there is not a peak around 45-50 kDa for MBP in the size exclusion experiments and only one for lambda phosphatase. The authors should clarify the methods and consider indicating in the text when MBP-Sae2 is used and when a cleaved Sae2 is used.

Answer: We performed all biochemical experiments and size exclusion chromatography with Sae2 without the MBP tag. The tag is cleaved during protein purification and is not present in our final protein preparations. After MBP tag cleavage, the protein is applied on NiNTA resin, and the MBP tag ends up in the flow through of the NiNTA column. We now clarify this point in the methods and include a reference to our previous publication where the purification is described in detail. "Briefly, Sae2 was expressed as a fusion with maltose binding protein (MBP), which was cleaved during purification and is absent in our final protein preparations as described²¹."

10. Is it possible that the large aggregates/oligomers of Sae2 that form are due to the higher concentrations of Sae2? Also, there is DNA present in Figure 1F and not in the size exclusion experiments. It is very likely that Sae2 whether phosphorylated or not will take on conformational changes when bound to DNA. The authors need to reconsider their conclusion on page 15 that pSae2 tetramers represent the active pSae2 species that optimally promote the Mre11 nuclease within the MRX complex. The authors did not perform the size exclusion chromatography with DNA present whereas the nuclease experiment obviously did. Therefore, the comparison between the DNA cleavage assay and the oligomeric state of Sae2 does not seem to be appropriate without further clarification.

Answer: We believe that protein concentration will likely affect the equilibrium between the oligomeric and multimeric species to a certain degree. Therefore, we performed size exclusion chromatography with consistent amounts of the Sae2 variants (~75 µg in 200 µl, as described in methods), so the results should not be affected by variations in protein concentration.

As suggested by the reviewer, when discussing the oligomeric state, we now note clearly that this refers to Sae2 that is DNA-free in solution.

11. *The use of AFM to estimate the formation of a tetramer is very nice.*

Answer: Thank you!

12. *Although qualitative, the authors use of TEM is also nice to see. The authors state that non-sized Sae2 was used in the TEM experiments which yielded mix populations of species of Sae2 to be visualized. These results are in agreement with the size exclusion experiments. That said, it is not clear why the authors did not evaluate the sized Sae2 variants. This would have clearly confirmed the which species of Sae2 is forming aggregates/multimers and oligomers.*

Answer: We aimed to obtain insights into the differences between the phosphorylated and non-phosphorylated Sae2 variants, and therefore did not size the sample before the TEM analysis. We used AFM to obtain more quantitative data, because of the better information on depth. Our aim is to obtain more detailed structural insights using cryo-EM in the future, but this appears to be very challenging.

13. *Did the authors visualize the different species of Sae2 on DNA?*

Answer: No, we have not, and we agree that this will be very important (with and without MRX). Please see our response above. We hope the reviewer will agree that this analysis is above the scope of the current manuscript, in particular as we do not think that Sae2 binding is critical for the MRX-Sae2 DNA clipping.

14. *It is possible that the phosphorylation status of Sae2 may have altered the way Sae2 interacts with the carbon film in the TEM. For instance it may have flattened out. To this end, the authors mentioned that the AFM images were corrected for flattening. This suggests that the different phosphorylation states of Sae2 may have altered the way Sae2 interacts with the freshly cleaved mica. The authors should comment on these issues more than is currently in the manuscript.*

Answer: We believe that phosphorylated Sae2 (or weakly phosphorylated Sae2) will be negatively charged and therefore unlikely to show extra-flattening on negatively charged grid surfaces. We add to the methods a note: "under conditions which create a negatively charged grid surface".

15. *In Figure 2 panel D, why is there not a lambda phosphatase band present? If the molecular weight markers are correct, it should be present on the gel as it is in every gel in the manuscript. Is the lambda phosphatase masked by the Sae2-delta169 variants?*

Answer: Indeed, we believe that the bands will co-migrate and cannot be separated.

16. *Figure 3, the quantitation in B is not in agreement with the respective panels in A. 17. Figure 3, the quantitation in G is not in agreement with the respective panel in F. 18. Figure 4, the quantitation in L is not in agreement with the respective panels in K.*

Answer: Generally, we observe some variability in the overall reaction efficiency between individual experiments. As noted above, sometimes we can detect MRX-only cleavage, which is particularly apparent when we use a very "hot" DNA substrate, corresponding up to ~8% cleavage. When we use the same substrate 1 month later, we may not detect this activity. This resulted in inconsistencies in quantitations. Therefore, we always remove the

background Mre11 activity so that we only score for effects of added Sae2, and the curves thus start at "0". By mistake, this was not included in methods - now this has been corrected, and we detail how quantitation has been performed. "Images were quantitated using ImageQuant software. First, DNA cleavage in MRX-only lanes (without Sae2) was removed as a background from all other lanes to score specifically for effects of Sae2. Subsequently, the Sae2-dependent DNA cleavage was calculated as products / (substrate + products) in each lane."

19. On page 9, the authors state that pSae2 that had been dephosphorylated on page 9, the authors state that pSae2 that had been dephosphorylated by lambda phosphatase can be partially activated upon phosphorylation by human CDK1/Cyclin B. Perhaps the authors included the wrong panel for F in figure 3, because the gel has no appreciable change in the cleavage pattern. This is in stark contrast to the graph in panel G. Which is correct?

Answer: Please see our response above. The experiments with CDK1/Cyclin B in particular showed activity in MRX only lane. We wonder whether this is due to phosphorylation of MRX, or a non-specific component of the reaction buffer. This cleavage has been removed from the quantitations to focus on the effects caused by Sae2. Irrespectively of this, we do observe a clear difference between phosphorylated and non-phosphorylated Sae2. The pattern i.e. cleavage positions do not change, but the cleavage efficiency is increased. Please note the ratio of products vs. substrate left: there is a clear difference between lanes 3-5 and 6-8. It may be that an alteration of image resolution and/or contrast change during pdf conversion might have masked this. Please consult the full-resolution image that has been submitted with the revised version of the manuscript.

20. Since the authors point out on page 15 that ATP hydrolysis by Rad50 is necessary for the MRX-Sae2 endonuclease, does unphosphorylated or phosphorylated Sae2 influence the ATPase activity of Rad50?

Answer: We are attempting to answer this question for a very long time, and so far unsuccessfully. We have indirect biochemical evidence (not shown here) that supports this conclusion, but we were not able to demonstrate a stimulation of ATP hydrolysis directly. Possibly, Sae2 might stimulate the productive ATP hydrolysis (and not overall ATP hydrolysis) of Rad50, which would be hard to detect.

Reviewer #3 (Remarks to the Author):

Sae2 is a DNA repair enzyme conserved across most species. Its major role is regulation of Mre11-Rad50-Xrs2 complex, a key multifunctional enzyme in DNA damage response. One of the major functions of MRX complex is in resection of 5' ends of DNA double strand breaks to form 3' ssDNA, essential process for recombination and DNA damage checkpoint. The authors of this manuscript focus on the control of MRX activity in resection by Sae2. Specifically they look at the role of Sae2 phosphorylation. They purified Sae2 from Sf9 cells in the presence of phosphatase inhibitors. Purified this way Sae2 was phosphorylated to much higher level than observed before and showed increased activities meaning higher stimulation of Mre11 nuclease activity. They provide convincing evidence that phosphorylation of Sae2 regulates oligomerization of Sae2. Hyperphosphorylated form of Sae2 forms specific oligomeric form (tetrameric) while nonphosphorylated

form is observed mostly as multimeric. The domains and phosphorylation sites needed for the formation of tetrameric form and the importance of tetrameric form are defined. The authors also carefully examine the function of Mec1 and Tel1 mediated phosphorylation and Cdk1 mediated phosphorylation of Sae2 in stimulating MRX activity and in formation of most active tetrameric form. They performed mass spec to identify additional phosphorylation sites and document that Sae2 phosphorylation at C-terminus is important for optimal activity of Sae2.

This is the most comprehensive study of Sae2 phosphorylation. It clarifies some important controversies (oligomerization, nuclease activity) and provides new mechanistic insights on how phosphorylation of Sae2 stimulates MRX activity. It also opens a lot of questions for the future studies on Sae2. Considering that MRX and Sae2 play central role in damage response and recombination this work deserves strong consideration. The manuscript is well written.

Minor questions/concerns that authors could discuss.

Is there a possibility that hyperphosphorylated form of Sae2 as purified from Sf9 cells is not the form that exist normally in yeast. Perhaps phosphatases eliminate most of the phosphate groups in cells or kinases in Sf9 cells have different specificity. It seems that besides Cdk1 and Tel1/Mec1, other kinases are involved as many sites tested here and shown to be important for optimal activity are not within the consensus motifs of these kinases. The authors should comment on this. Is there any order of phosphorylation events (one phosphorylation is needed to observe another one?). Are all phosphorylation events controlled in cell cycle? It seems that Cdk1 sites within Sae2 are not needed for other Sae2 phosphorylation events, so how other phosphorylation events are controlled in cell cycle?

Answer: We agree with the reviewer. We discuss this in the revised text: working with a protein phosphorylated in a non-cognate system obviously has its caveats. "... as our protein was expressed in insect cells, we cannot exclude that some of these sites are not modified in yeast, or are rapidly dephosphorylated in cells." That said, we are less concerned about this because phosphorylation results in a gain of activity. Also, we attempted to address the specific sites in Sae2 using both in vitro and cellular systems, where possible. We believe that other kinases (e.g. Cdc5 or DDK) may likely be involved. Based on our data, we cannot comment on the order or cell cycle-dependence of phosphorylation events. Some of that has been addressed by T. Paull (ref. Fu et al, MCB, 2014) and X. Wu with CtIP (Wang et al., Plos Genetics, 2013). The CDK site of S267 is clearly regulated in a cell cycle dependent manner, and it is likely that it affects other phosphorylation sites.

Other points

Page 9: "The ~5-fold decrease in MRX-dependent DNA cleavage upon dephosphorylation of pSae2 S267E demonstrated that other sites in pSae2, in addition to S267, must be phosphorylated for its optimal stimulation of the MRX endonuclease." It's a bit too strong conclusion. The alternative view is that phosphomimetic form of Sae2 only partially restores the activity of Sae2. Phosphomimetic forms of proteins often have intermediate level of activity in cells. In general the authors should consider this possibility when testing pseudo phosphorylated forms of Sae2.

Answer: The S267E mutant is hypomorphic *in vivo*, as reviewer points out (seen also here, but already apparent in the original paper by Huertas et al., 2008). We observed that the S267E mutation resulted in defects *in vitro* when used in within the C-terminal fragment, but had surprisingly little effect within the full-length protein (compared to phosphorylated wild type).

However, in the particular experiment that the reviewer refers to, we compared phosphorylated Sae2 S267E (this had the phosphomimetic mutation, plus was phosphorylated at other sites) versus the same mutant treated with lambda PP, where the other phosphorylation modifications have been eliminated. In this context, we believe that our conclusion stands. We however agree that lambda treated S267E will be probably less active than a variant that contains phosphorylated serine but no other modifications.

Abstract

"The lack of this interaction explains the phenotype of rad50S mutants defective in the processing of Spo11-bound DNA ends during meiotic recombination."

It would be great to introduce rad50S better or rephrase the statement for general audience.

Answer: Due to the word limit in abstract, we include a more detailed description later in the introduction: "The resection and resulting recombinational repair of meiotic DSBs absolutely require Sae2 and MRX and their orthologs, because Spo11 remains covalently bound to the break ends and needs to be removed by MRX and Sae2¹⁴⁻¹⁷. This stands in contrast to resection in yeast vegetative cells, which can be partially MRX-Sae2 independent^{18,19}. To this point, Rad50 separation of function alleles (*rad50S*) have been found, which prevent Spo11 removal in meiosis, but are less defective in vegetative cells."

In the model on Figure 7, It would be good to add the names of kinases involved.

Answer: We think focusing more on the kinases is a good idea, but decided to include the discussion of the respective (and putative) kinases in Discussion. First, we do not want to complicate the cartoon with too much text. Second, as the identity of all the kinases is not clear, we think placing this text in Discussion is more appropriate.

"Phosphorylation of the conserved CDK site of Sae2 at S267 is critical for its resection function *in vitro* and *in vivo*⁶, but there is a need for the modification of a number of additional sites that play a supporting role. This may include other CDK sites, Mec1/Tel1 sites, or other kinases such as Cdc5 or the Dbf4-dependent kinase (DDK), whose function in Sae2 regulation has however not yet been defined."

Page 12

*"For subsequent analysis, we selected 22 putative pSae2 phosphorylation sites based on our mass-spectrometry analysis and previous work; 15 of those pSae2 sites had been previously found to be modified *in vivo*"*

Many of these published sites are coming from high throughput studies and were not confirmed. It's a bit overstatement to say these all sites are phosphorylated in cells. It would be good to add information how these sites were identified.

Answer: Thank you. We now include a note that "13 of these sites were identified in Flag-Sae2 pulldown from yeast cells⁸, and thus very likely to be modified *in vivo*."

Next, we write: "However, as our protein was expressed in insect cells, we cannot exclude that some of the non-overlapping sites are not modified in yeast, or are rapidly dephosphorylated in cells."

Then in Discussion, we add: "Although we found 22 sites that are likely to be modified in the population of recombinant pSae2, most individual polypeptides will contain only a fraction of those modifications."

REVIEWERS' COMMENTS:

Reviewer #1 (Remarks to the Author):

The authors have done an excellent job revising the manuscript according to my previous critique.

The revised paper helps resolve outstanding questions about MRX-Sae2 function and makes an important contribution in studies of DNA resection in eukaryotes.

Reviewer #2 (Remarks to the Author):

The authors have provided adequate and acceptable revisions that address my concerns. The authors have adequately and acceptably answered all my questions. My enthusiasm for this manuscript is very high. The authors have done an exceptional job in this study. The understanding of Sae2 regulation of MRX has substantially increased above what was previously known. I am still blown away by the inclusion of the pSae2-22E variant in this manuscript. This manuscript is very much suited for publication in Nature Communications.

Reviewer #3 (Remarks to the Author):

The revised manuscript is improved. It addresses most of the questions raised by this reviewer. This work provides new insights into regulation of DSB ends resection.